# Clathrin-independent pathways do not contribute significantly to endocytic flux

**Vassilis Bitsikas[1], Ivan R Corrêa Jr[2], Benjamin J Nichols[1]\***

[1]Department of Cell Biology, Medical Research Council Laboratory of Molecular Biology, Cambridge, United Kingdom; [2]New England Biolabs, Inc., Ipswich, United States

**Abstract** Several different endocytic pathways have been proposed to function in mammalian cells. Clathrin-coated pits are well defined, but the identity, mechanism and function of alternative pathways have been controversial. Here we apply universal chemical labelling of plasma membrane proteins to define all primary endocytic vesicles, and labelling of specific proteins with a reducible SNAP-tag substrate. These approaches provide high temporal resolution and stringent discrimination between surface-connected and intracellular membranes. We find that at least 95% of the earliest detectable endocytic vesicles arise from clathrin-coated pits. GPI-anchored proteins, candidate cargoes for alternate pathways, are also found to enter the cell predominantly via coated pits. Experiments employing a mutated clathrin adaptor reveal distinct mechanisms for sorting into coated pits, and thereby explain differential effects on the uptake of transferrin and GPI-anchored proteins. These data call for a revision of models for the activity and diversity of endocytic pathways in mammalian cells.

**\*For correspondence:** ben@mrc-lmb.cam.ac.uk

## Introduction

Endocytosis has central roles in many cell biological processes (*Le Roy and Wrana, 2005*; *Doherty and McMahon, 2009*). Since the mid-nineties evidence has accumulated to suggest that mammalian cells utilise additional endocytic mechanisms beyond clathrin-coated pits (*Sandvig and van Deurs, 1994*; *Mayor and Pagano, 2007*; *Sandvig et al., 2008*). Whilst the molecular detail of how clathrin-coated pits work is understood in ever-increasing detail (*Kirchhausen et al., 2014*), similarly complete mechanistic descriptions of how endocytosis may take place outside of clathrin-coated pits are lacking.

One central difficulty in defining clathrin-independent endocytic pathways has been the paucity of rigorously validated endocytic markers and pathway-specific cargoes. Dissection of clathrin-mediated endocytosis benefited greatly from signature cargoes such as the transferrin receptor, which are efficiently concentrated in the forming endocytic vesicle, and from the fact that the forming vesicle is marked specifically in space and time by transient assemblies of clathrin, adaptors and associated proteins (*Pearse, 1982*; *Dautry-Varsat et al., 1983*; *Doxsey et al., 1987*; *Ehrlich et al., 2004*). By contrast, clathrin independent endocytosis has largely been defined by morphological criteria, and by the persistent uptake of cargoes that may utilise multiple pathways following perturbation of the clathrin machinery (*Saslowsky et al., 2010*; *Engel et al., 2011*; *Cho et al., 2012*).

Endocytic structures that can be defined morphologically include macropinosomes, which can readily be resolved by light microscopy (*Liberali et al., 2008*), and caveolae which are distinctive in electron micrographs (*Parton and del Pozo, 2013*). The extent to which caveolae are involved in endocytosis is, however, controversial (*Parton and Howes, 2010*). Morphology also forms a large part of the definition of more recently characterised endocytic membranes termed CLICs, for clathrin-independent carriers (*Kirkham et al., 2005*; *Howes et al., 2010*).

**eLife digest** Cells are enclosed by a 'plasma membrane' that allows nutrients and certain small molecules to move in and out of cells. Larger molecules—such as proteins—are carried into cells through a process known as endocytosis, where part of the plasma membrane engulfs the molecule and transports it through the cell inside a bubble-like compartment called a vesicle.

There may be several different ways by which endocytosis can occur. The most common method involves a protein known as clathrin, which coats part of the plasma membrane on the side facing the inside of the cell. This causes the membrane to deform into a pit. The pit grows around, and eventually completely surrounds, the molecule to be transported, at which point the clathrin-coated membrane pinches off from the rest of the plasma membrane to form a vesicle.

Other forms of endocytosis do not need clathrin to form vesicles, and so are collectively known as clathrin-independent endocytosis. However, the details of how these other types of endocytosis work and how important they are for moving molecules into the cell remain unclear. This is partly because it is difficult to identify particular types of endocytosis. Previous attempts to do this have involved trying to identify molecules that are specifically and solely associated with that type of endocytosis, and using these to track the vesicle. However, few—if any—such molecules are known for clathrin-independent methods of endocytosis. Another approach is to inhibit the formation of clathrin-coated pits and study those molecules that are still taken into cells. The problem here is that incomplete inhibition can make interpreting the results difficult. Furthermore, complete inhibition of an important process like clathrin-dependent endocytosis is likely to have severe effects on many other aspects of cell function.

Bitsikas et al. have developed a new method that allows a vesicle to be identified—regardless of how it forms—in cells that have not been treated with inhibitors. This method involves labelling proteins in the plasma membrane with a chemical that allows them to be traced, and so shows when they are included in vesicle membranes. Importantly, this new method can provide very accurate information as to whether or not proteins have been included in vesicles, and this may provide advantages over previous approaches.

Bitsikas et al. selected a group of proteins that are thought to only enter cells in a clathrin-independent manner, but unexpectedly found that these proteins predominantly enter cells through clathrin-coated vesicles. Further analysis revealed that approximately 95% of all molecules that enter cells by endocytosis are taken up via clathrin-coated endocytosis. Therefore, clathrin-independent endocytosis does not make a significant contribution to the transport of large molecules into cells.

These results are at odds with current thinking in the field. Future work could reveal whether the techniques applied by Bitsikas et al. detect more active clathrin-independent endocytosis in special situations, for example during cell migration, or in specific cell types.

GPI-anchored proteins have been extensively studied as potential cargoes for clathrin-independent endocytosis (*Mayor and Pagano, 2007*; *Johannes and Mayor, 2010*). The apparent presence of these proteins in endosomes devoid of transferrin, and uptake in the presence of inhibitors of clathrin-coated pits, provides evidence for GPI-enriched endosomal compartments (GEECs) that are fed from the cell surface independently from coated pits (*Sabharanjak et al., 2002*; *Kumari and Mayor, 2008*; *Bhagatji et al., 2009*). It is not clear, however, that GPI-anchored proteins are ever highly concentrated in nascent endocytic vesicles in a manner analogous to transferrin receptor, so they may enter the cell via multiple mechanisms (*Mayor and Riezman, 2004*; *Sharma et al., 2004*; *Bhagatji et al., 2009*; *Johannes and Mayor, 2010*). The extent to which other types of cargo associated with clathrin-independent endocytosis, including glycosphingolipid-binding bacterial toxins as well as various viruses, are efficiently sorted during uptake is similarly unclear (*Romer et al., 2007*; *Ewers et al., 2010*; *Johannes and Mayor, 2010*; *Saslowsky et al., 2010*; *Cho et al., 2012*; *Lakshminarayan et al., 2014*).

In the absence of demonstrably specific cargoes, much of the literature on clathrin-independent endocytosis relies on the use of overexpressed mutant proteins to perturb clathrin function, and observation of differential effects on the uptake of transferrin and potential clathrin-independent

cargoes. Dominant negative mutants used in this manner include the C-terminal clathrin-binding domain of AP180/CALM (*Ford et al., 2001*), and the K44A mutant of dynamin, which renders this GTPase involved in the scission of clathrin coated pits enzymatically inactive (*van der Bliek et al., 1993*; *Damke et al., 1995*). Additionally, differential effects on endocytosis can be observed using overexpression of inactive forms of small GTPases such as ARF6, ARF1 or cdc42 (*D'Souza-Schorey et al., 1995*; *Palacios et al., 2002*; *Sabharanjak et al., 2002*; *Naslavsky et al., 2004*; *Kumari and Mayor, 2008*). Blocking one type of endocytosis may up-regulate alternative mechanisms, over-expression of mutant proteins may induce non-physiological cellular responses, and small GTPases may, via different sets of effectors, directly or indirectly control the activity of multiple endocytic pathways. Plainly, these types of experiment need to be interpreted carefully.

Ideally, different types of endocytosis would be defined by the presence of specific molecular determinants analogous to clathrin or adaptor proteins in the case of clathrin coated pits (*Kirchhausen et al., 2014*). Candidates for such determinants include caveolin and cavin proteins which make caveolae (*Ludwig et al., 2013*; *Parton and del Pozo, 2013*), flotillin proteins which define specific plasma membrane microdomains potentially involved in endocytosis (*Glebov et al., 2006*), and the protein GRAF1 that may be important for the formation of CLIC/GEEC endosomes (*Lundmark et al., 2008*). The precise mechanisms by which these proteins are involved in endocytosis remain to be fully understood. Both flotillins and caveolins plus cavins form protein assemblies that are stable over time, and thus can not define endocytic events temporally in the way in which coordinated assembly and disassembly of the clathrin machinery can (*Glebov et al., 2006*; *Frick et al., 2007*; *Taylor et al., 2011*; *Gambin et al., 2013*; *Ludwig et al., 2013*).

Thus, although several non-clathrin endocytic pathways have garnered varying degrees of supportive evidence, none has been unambiguously established in molecular and functional terms. Furthermore, the relative contributions of the multiple putative pathways to overall endocytic flux has been unclear. To resolve some of these uncertainties, ideally one would require a means to examine endocytosis in unperturbed cells in a global manner. This would allow simultaneous evaluation of a large number of cargoes and their relationship with clathrin and putative non-clathrin markers. In this study we apply a combination of new and established methods that satisfy these requirements, and thereby provide a systematic and quantitative analysis of total endocytic protein flux in cultured mammalian cells.

## Results

### Experimental strategy and assay validation

We sought to establish endocytic assays and protein labelling strategies to satisfy four main requirements: (1) achieve very high topological specificity in discrimination between endocytosed and extracellular protein, (2) provide a means to analyse all, or nearly all, surface proteins simultaneously, (3) provide high signal to noise, and thereby high temporal resolution for detection of primary endocytic vesicles, and (4) allow a means to follow uptake of specific cargoes.

Biotinylation of extracellular free amine groups with the small, monovalent label sulfo-NHS-SS-biotin offered a way to satisfy the first three of these requirements (*Le Bivic et al., 1990*). The biotin moiety can be removed from labelled proteins by reduction of the disulfide bond with membrane-impermeant sodium 2-mercaptoethanesulfonate (MESNa) at 4°C, so this approach provides a powerful way to detect internalisation of surface proteins (*Bretscher and Lutter, 1988*; *Schmid and Smythe, 1991*). We assessed the efficiency of MESNa treatment in removing biotin from sulfo-NHS-SS-biotin-labelled proteins, and compared this with two widely used alternatives, washing at low pH to remove antibodies bound to the outside of the cell and cleavage of extracellular GPI-anchors with PI-PLC (*Sabharanjak et al., 2002*). MESNa treatment could remove over 99.9% of biotin from cells labelled at 4°C (*Figure 1A*). Washes at pH3 to remove bound antibody, or PI-PLC to cleave GPI-anchors, were around two orders of magnitude less efficient, removing up to 50% and 90% of antibody bound to the GPI-anchored protein CD59 respectively (*Figure 1B*) (*Davies et al., 1989*). Reducible biotin and MESNa therefore offer a highly accurate way of assaying protein internalisation.

To detect internalised protein in confocal images, we labelled HeLa cells with sulfo-NHS-SS-biotin, allowed internalisation for defined periods of time, removed extracellular biotin by reduction with MESNa, and used fluorescent streptavidin to detect intracellular biotin. All subsequent experiments are also in HeLa cells unless otherwise stated. Control experiments comparing incubation at 4°C with

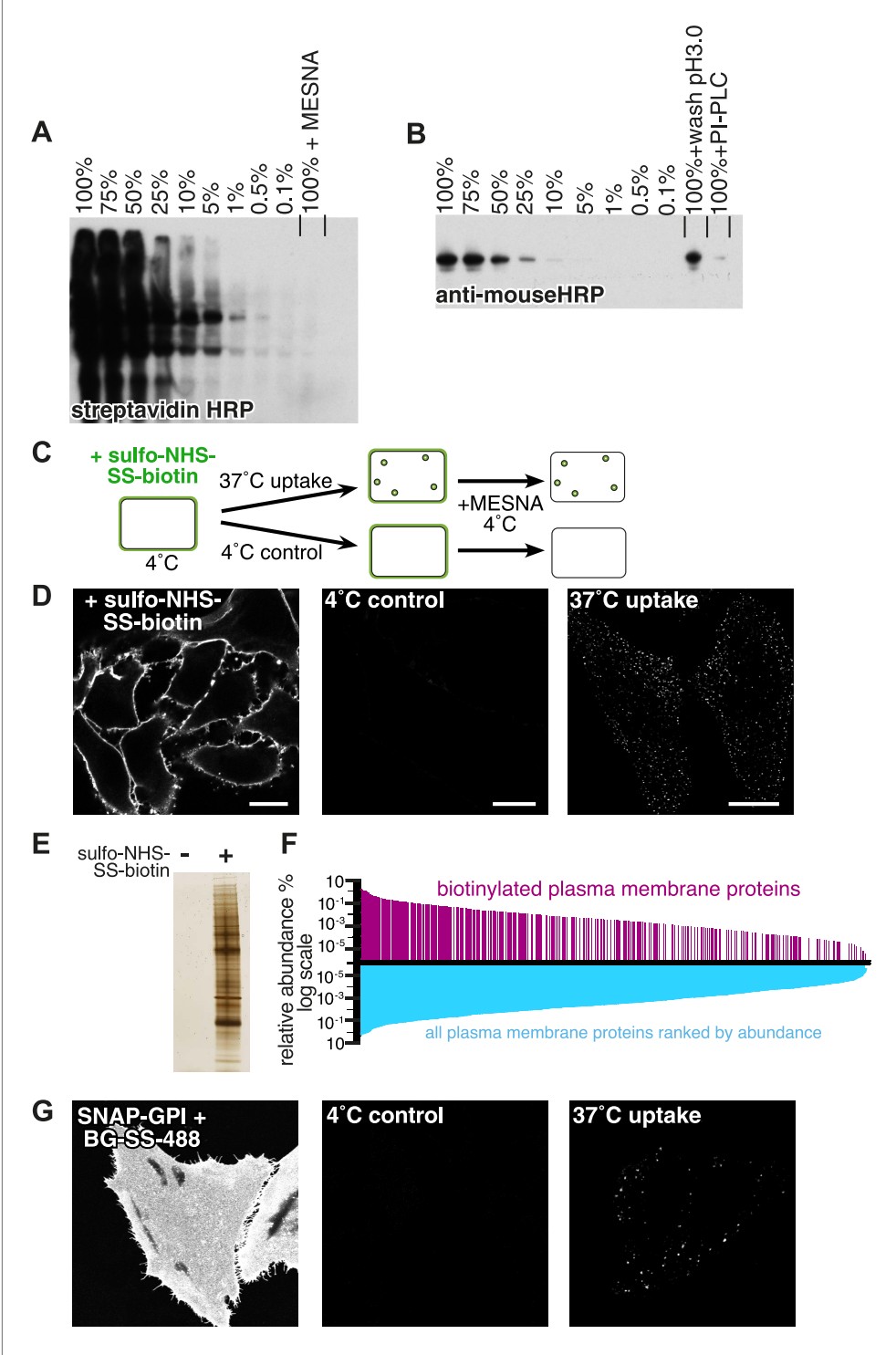

**Figure 1**. Experimental strategy and assay validation. (**A**) Following surface biotinylation with sulfo-NHS-SS-biotin, HeLa cell lysate was serially diluted with non-biotinylated control cell lysate, and the amount of labelling was detected using streptavidin-HRP after SDS-PAGE and blotting. Incubation with 100 mM MESNa prior to cell lysis was enough to remove more than 99.9% of the initial signal. (**B**) Lysate from cells labelled with anti-CD59 antibody was serially diluted with non-labelled control cell lysate, and the amount of labelling was detected using an anti-mouse HRP antibody. Acid wash removed around 50% of the initial surface signal. PI-PLC, which cleaves the GPI-anchor in CD59, removed up to 90% of the initial signal. (**C**) Cartoon to illustrate endocytosis assay. Cell surface

*Figure 1. Continued on next page*

*Figure 1. Continued*

proteins were labelled with sulfo-NHS-SS-biotin at 4°C, then the reaction was quenched and cells were rapidly transferred to 37°C to allow endocytosis. Control cells were kept at 4°C. After defined time for endocytosis, cells were rapidly returned to 4°C, and surface-exposed biotin was removed using the membrane impermeable reducing agent MESNa. Biotin label was detected using fluorescent streptavidin, after fixation and permeabilisation. (**D**) Confocal images from control experiment to demonstrate total surface labelling with sulfo-NHS-SS-biotin, negative control, and labelling of endocytic vesicles, as illustrated in **C**. Bars are 20 μm. (**E**) Silver stained SDS-PAGE gel following surface biotinylation with sulfo-NHS-SS-biotin and streptavidin pull-down. Non-biotinylated cells provided a negative control. (**F**) Surface biotinylation labels the full range of plasma membrane proteins ranked according to their relative abundance in the plasma membrane. The relative abundance of plasma membrane proteins in HeLa cells was calculated based on a previously published study (*Kulak et al., 2014*). These are represented graphically ranked by abundance in blue, while proteins were detected in our experiments are shown in magenta. The data are listed in *Figure 1—source data 1*. (**G**) Confocal images of HeLa cells stably expressing SNAP-GPI. Cells labelled at 4°C with BG-SS-488. Control cells were treated with MESNa to remove external fluorophore after incubation only at 4°C. Warming to 37°C for 90 s before MESNa treatment at 4°C allows identification of endocytic vesicles.

The following source data and figure supplement is available for figure 1:

**Source data 1**. Plasma membrane proteins identified by mass spectrometry.

**Figure supplement 1**. Removal of extracellular fluorophore from BG-SS-fluorophore labelled SNAP-tag by reduction with MESNa is highly efficient.

---

incubation for defined periods at 37°C confirmed that there was no specific signal detected unless internalisation at 37°C was allowed to take place (*Figure 1C*, *Figure 1D*). Endocytic vesicles could be observed after as little as 20 s of internalisation. This is a significant improvement in time resolution, and hence in our confidence that the detected vesicles have just budded from the plasma membrane (*Sabharanjak et al., 2002*; *Kirkham et al., 2005*; *Glebov et al., 2006*).

In order to interpret uptake of proteins labelled with sulfo-NHS-SS-biotin with confidence it was necessary to confirm that the label reacts with a broad range of representative proteins. Following surface biotinylation, proteins were precipitated using streptavin-agarose beads, and eluted from the beads by reduction with 100 mM DTT. Silver staining revealed that biotinylated proteins were precipitated with high specificity (*Figure 1E*). LC-MS identified the full list of precipitated proteins, and this list was compared with a quantitative analysis of the HeLa cell proteome (*Kulak et al., 2014*). Abundant plasma membrane proteins had higher chances of being detected. Nevertheless, a wide range of rare proteins (some with estimated copy number < 100 per cell) was detected, assuring us that we achieved good coverage of the total plasma membrane protein complement. (*Figure 1F*, *Figure 1—source data 1*). Therefore chemical labelling with sulfo-NHS-SS-biotin does indeed provide a way to follow endocytosis of nearly all plasma membrane proteins simultaneously.

Reduction of extracellular disulphide bonds with membrane impermeant reducing agents like MESNa evidently provides a good way to discriminate between intra- and extra-cellular substrates. We used the genetically encoded SNAP-tag to apply this approach to specific cargo proteins (*Gautier et al., 2008*; *Cole and Donaldson, 2012*; *Correa, 2014*). The SNAP moiety was labelled with a new membrane-impermeant, reducible, fluorescent SNAP-labelling reagent, benzylguanine-SS-fluorophore [where the fluorophore can be either atto-488 (BG-SS-488) or BODIPY (BG-SS-549)]. The disulfide bond linking benzylguanine to a fluorophore can be reduced with MESNa in the same way as that in sulfo-NHS-SS-biotin. Control experiments using a minimal GPI-anchored SNAP-tag construct as a model plasma membrane protein showed that BG-SS-fluorophore allows detection of internalised SNAP tag with very high efficiency and low background, and provides an improved way of discriminating between internal and external pools of tagged protein (*Figure 1G*, *Figure 1—figure supplement 1*).

## Biotinylated plasma membrane proteins are predominantly internalised via clathrin coated pits

Global labelling of primary endocytic vesicles was achieved by incubation of cells with sulfo-NHS-SS-biotin at 4°C, warming to 37°C by rapid buffer exchange, and allowing uptake for 20 s. Endocytic vesicles detected after MESNa treatment were small puncta (*Figure 2A*). Approximately 2% of the cells also contained distinctively larger macropinosomes over 500 nm in diameter (*Figure 2—figure supplement 1*).

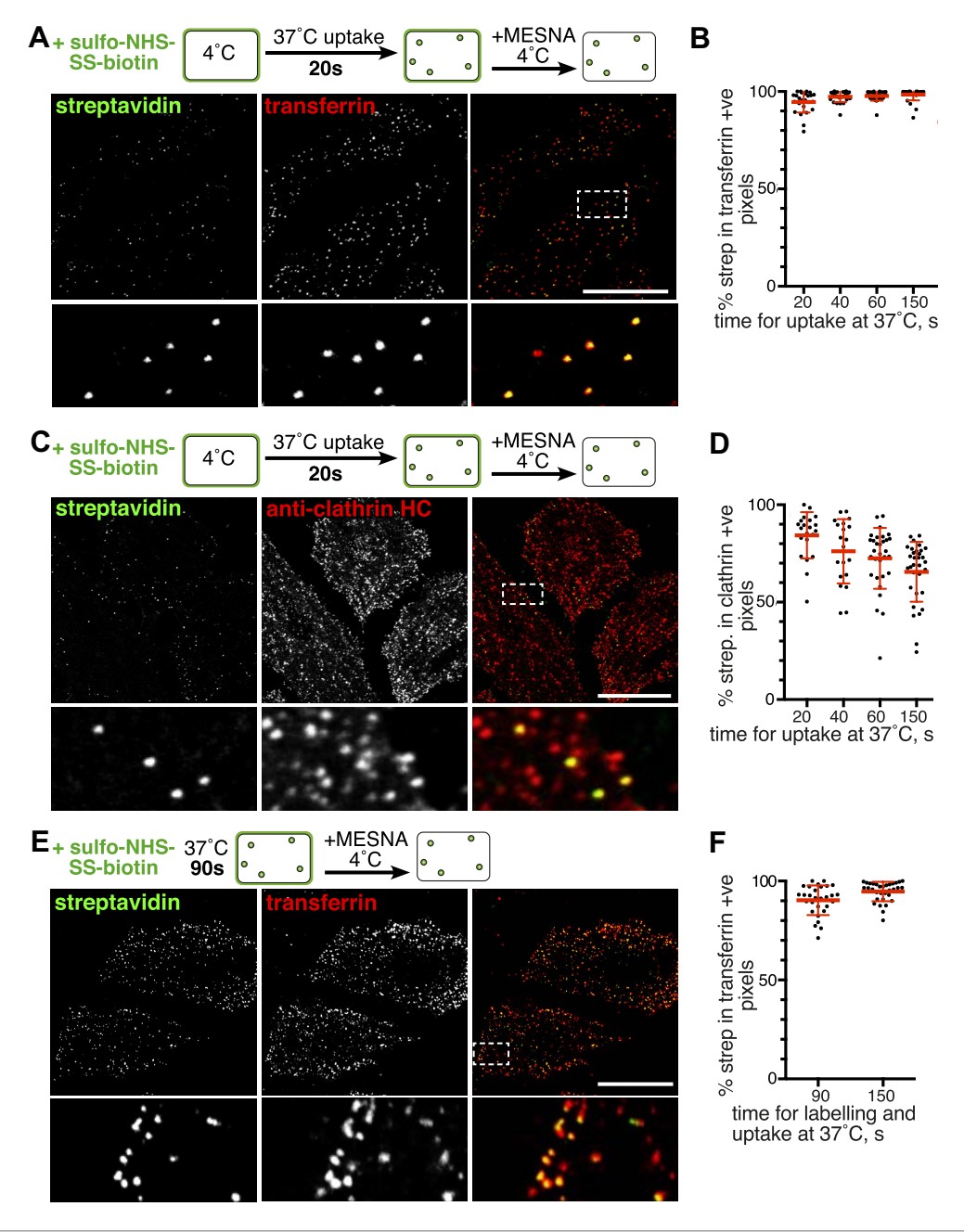

**Figure 2**. Over 95% of total endocytosed protein co-localises with markers for clathrin-mediated endocytosis. (**A**) Confocal images of co-internalisation of all membrane proteins, labelled at 4°C with sulfo-NHS-SS-biotin, and transferrin-546. Internalisation was for 20 s at 37°C. Biotin was detected with streptavidin-488. Note that labelling with biotin and transferrin at 4°C was carried out consecutively, so transferrin was not biotinylated. The presence of external transferrin in nascent coated pits explains transferrin-positive, strepatavidin-negative puncta (see *Figure 2— figure supplement 2*). Zoomed in area in the lower panels is indicated with a box. Bar is 20 µm. (**B**) Quantification of proportion of internalised protein, detected as in **A**, that co-localises with transferrin. Bars are mean and SD, the data are from one experiment that was repeated three times with the same overall result. (**C**) Confocal images of total endocytosed membrane protein, labelled at 4°C with sulfo-NHS-SS-biotin as shown, and indirect immunofluorescence staining for clathrin heavy chain. Internalisation was for 20 s at 37°C. Biotin was detected with streptavidin-488. (**D**) Quantification of proportion of total internalised protein that co-localises with clathrin, at the times of internalisation indicated. Bars are mean and SD, the data are from one experiment that was repeated three times with the same result. (**E**) Confocal images of co-internalisation of total membrane protein and transferrin-546, with 90 s at 37°C for

*Figure 2. Continued on next page*

*Figure 2. Continued*

labelling and uptake as illustrated in the cartoon. Bar is 20 µm. (**F**) Quantification of proportion of internalised protein, detected as in **E**, that co-localises with transferrin, at the times for continuous labelling and internalisation indicated. Bars are mean and SD, the data are from one experiment that was repeated three times with the same result.

The following figure supplements are available for figure 2:

**Figure supplement 1**. Macropinosomes are readily identified by labelling with sulfo-NHS-SS-biotin.

**Figure supplement 2**. Surface-bound transferrin is highly concentrated within clathrin-coated pits.

**Figure supplement 3**. Quantification of percent co-localisation.

**Figure supplement 4**. Co-localisation between internalised sulfo-NHS-SS-biotin and transferrin after labelling at 4°C and 90 s of internalisation at 37°C.

**Figure supplement 5**. Co-localisation between internalised sulfo-NHS-SS-biotin and clathrin after 20 s and 60 s of internalisation at 37°C.

**Figure supplement 6**. Total endocytosed protein and transferrin co-localise after 90 s uptake in Cos7 and RPE cells.

**Figure supplement 7**. Absence of membrane-positive, transferrin-negative vesicles.

In order to determine the identity of the endocytic vesicles we carried out co-internalisation experiments with the archetypical high-affinity cargo for clathrin-coated pits, transferrin (*Figure 2A*, *Figure 2—figure supplement 2*) (*Schmid and Smythe, 1991*; *Hansen et al., 1992*). The great majority of endocytic vesicles contained transferrin, and therefore are likely to have arisen from clathrin-coated pits. Uptake of transferrin was solely due to binding to transferrin receptor, as large, fluid-filled macropinosomes were completely devoid of transferrin (*Figure 2—figure supplement 1*). Residual surface-bound transferrin was detected as puncta, and co-localised with clathrin, implying concentration in nascent coated pits (*Figure 2—figure supplement 2*). We used unbiased and semi-automated quantification, involving application of a mask derived from the image of fluorescent transferrin, to determine the proportion of biotin-positive pixels that also contained transferrin in multiple cell images (*Figure 2—figure supplement 3*). This revealed that over 95% of the total internalised biotin after 20 s uptake was present in transferrin-positive vesicles (*Figure 2B*). Co-localisation remained constant when times of internalisation up to 150 s were assayed (*Figure 2B*, *Figure 2—figure supplement 4*). Additionally, after 20 s uptake over 80% of biotin-positive endocytic vesicles co-localised with clathrin in small puncta, defining them as clathrin-coated vesicles (*Figure 2C*, *Figure 2D*, *Figure 2—figure supplement 5*). The proportion co-localising with clathrin fell rapidly with longer periods of uptake, as one would expect due to uncoating of primary vesicles (*Figure 2D*, *Figure 2—figure supplement 5*) (*Kirchhausen et al., 2014*).

The experiments described above depended on labelling cells at 4°C before rapid warming to 37°C to permit endocytosis. This improved signal to noise ratio and thereby improved temporal resolution, but at the same time cooling and rapid warming could have an effect on the endocytic machinery (*Boucrot et al., 2010*). We sought to perform the endocytic assay in unperturbed cells, under more physiological conditions, by omitting the pre-incubation step at 4°C. Cells were labelled with sulfo-NHS-SS-biotin and transferrin at 37°C (*Figure 2E*). The earliest time after initiation of labelling at which endocytic vesicles could be reliably detected was 90 s, presumably because of the time taken for sufficient reaction of the NHS ester with primary amines at the cell surface. Consistent with the results when labelling was carried out at 4°C, nearly all of the internalised protein labelled with biotin was indeed present in transferrin-positive vesicles (*Figure 2E*, *Figure 2F*). Another concern was that HeLa cells could be in some way atypical, so we repeated the same experiments in Cos7 and RPE cells. Again, there was near-complete co-localisation between endocytosed sulfo-NHS-SS-biotin and transferrin after 90 s uptake at 37°C (*Figure 2—figure supplement 6*).

In order to verify that transferrin-negative endocytic vesicles are very rare we used a different labelling approach. We loaded HeLa cells with the amphiphilic styryl membrane dye FM1-43FX (*Diefenbach et al., 1999*), and allowed co-internalisation with transferrin for 90 s at 37°C. Extraction of the dye with label-free medium reversed plasma membrane labelling and provided a means of identifying FM1-43FX-positive intracellular vesicles. Although this may not provide such stringent topological discrimination between intracellular vesicles and surface membrane as the sulfo-NHS-SS-biotin plus MESNa approach, there was a very high degree of co-localisation between FM1-43FX-positive puncta and transferrin (*Figure 2—figure supplement 7*). Quantification revealed that 90% of the FM1-43FX signal present in intracellular vesicles was also present in transferrin-positive pixels (*Figure 2—figure supplement 7*).

Confocal z-stacks and volume rendering were used to allow analysis at the level of individual primary endocytic vesicles. (*Figure 3A*, *Figure 3—figure supplement 1A*). Biotin-positive endocytic vesicles were identified as 3D objects using the Imaris software, and the transferrin cargo load was calculated based on the mean transferrin intensity within each vesicle (*Figure 3A*, *Figure 3B*, *Video 1*). In order to account for background signal generated by experimental noise or random overlap, the transferrin channel was offset by 500 nm from its correct register, and transferrin intensity in the same objects was sampled for a second time. (*Figure 3B*). The 95th percentile of this background intensity distribution was used as a cut-off to define transferrin-negative endocytic vesicles (*Figure 3B*). Applying this conservative criterion, after 90 s continuous labelling and uptake 96% of 2350 vesicles contained transferrin, and after labelling at 4°C and uptake for 20 s 92% of 2387 vesicles contained transferrin (*Figure 3B*). A single Gaussian curve could describe the transferrin intensity distribution within most identified vesicles, consistent with stochastic incorporation of transferrin receptors into forming clathrin-coated pits (*Collinet et al., 2010*). A small population of transferrin-negative objects could be seen as a peak at the lowest end of the intensity distribution, showing that potential transferrin-negative endocytic vesicles, where they exist, can be detected by our method (*Figure 3B*). We observed no correlation between biotin intensity and the probability of that vesicle not containing transferrin, arguing against the possibility of a morphologically distinctive class of transferrin-negative endocytic vesicle (*Figure 3—figure supplement 1B*). We also carried out experiments to test the possibility that the glycosphingolipid-binding B-subunit of cholera toxin (CTB), which has been extensively used as a marker for clathrin-independent endocytosis (*Henley et al., 1998*; *Sandvig et al., 2004*; *Kirkham et al., 2005*), induces the formation of transferrin-negative endocytic vesicles. Biotin-positive vesicles were identified as objects in transferrin-labeled cells as described above, with and without addition of CTB. The transferrin load in vesicles in control and CTB treated cells was the same (*Figure 3—figure supplement 2*).

Both quantitative analysis of single confocal sections using pixel masks (*Figure 2*), and object-based quantification of all primary endocytic vesicles in 3D reconstructions (*Figure 3*), argue strongly that around 95% of primary endocytic vesicles are positive for the best characterised cargo of clathrin-coated pits, the transferrin receptor (*Kirchhausen et al., 2014*). The simplest hypothesis arising from these observations is that essentially all proteins taken up by the cell are internalised along with transferrin via clathrin-coated pits. If this is the case, then ablation of coated pit activity should effectively block total endocytosis. Overexpression of the C-terminal clathrin-binding domain of AP180/CALM (AP180-C) provides an efficient means of blocking formation of coated pits (*Ford et al., 2001*). In cells overexpressing AP180-C, transferrin uptake and endocytosis of total biotinylated protein were both efficiently blocked, with total protein uptake being reduced to less than 5% of control levels (*Figure 3C* and *Figure 3E*). Overexpression of the K44A mutant of dynamin blocks budding of clathrin-coated pits (*van der Bliek et al., 1993*; *Damke et al., 1995*). In cells overexpressing dynamin 2 K44A there was efficient reduction of both transferrin and total biotinylated protein uptake to less than 5% of control levels (*Figure 3D* and *Figure 3F*). Additionally, we noted that expression of very high levels of the dynamin mutant induced endocytosis of total biotinylated protein in both macropinosomes and smaller vesicular structures (see below). We conclude that formation of the primary vesicles that contribute the large majority of endocytic flux in unperturbed cells is blocked by loss of clathrin or dynamin function.

## Quantitative proteomics shows that perturbation of the formation of coated pits alters the protein composition of the plasma membrane

If plasma membrane proteins are predominantly endocytosed via clathrin coated pits then one would predict that they should accumulate in the plasma membrane when coated pits are not functional. Proteomic analysis of surface-biotinylated plasma membrane proteins offered a way to test this hypothesis

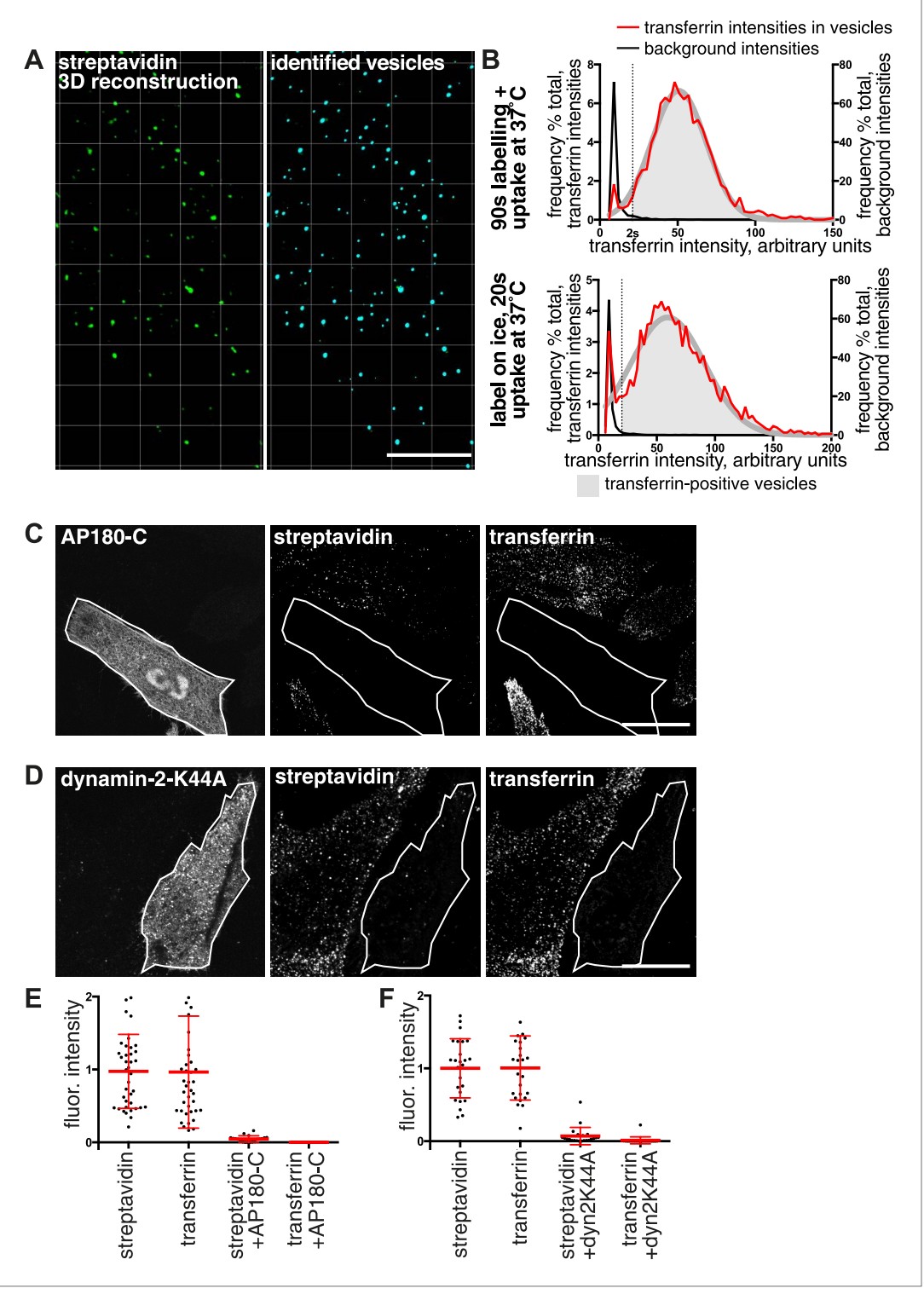

**Figure 3**. Over 95% of total endocytosed protein enters the cell via clathrin-coated pits. (**A**) 3D projection of cell volumes following interalisation of sulfo-NHS-SS-biotin for 90 s at 37°C. Streptavidin-488 fluorescence is shown in the left panel, vesicle objects recognised with Imaris software from the streptavidin signal are shown in the right panel. Bar 5 μm. (**B**) Analysis of the transferrin cargo load of endocytic vesicle objects identified as in **A**, after 20 s and 90 s of uptake as shown. Frequency distribution of mean transferrin intensity in individual vesicles is shown as the red line. Frequency distribution of transferrin intensities for the same vesicles after offsetting the transferrin

*Figure 3. Continued on next page*

*Figure 3. Continued*

channel by 500 nm provides a set of background intensities, shown as a black line and not plotted to the same y-axis scale. Cut-offs are shown as dotted lines and correspond to the 95 percentile for the offset values. The distribution of transferrin intensity, in the majority of endosomes, can be described by a Gaussian distribution (dark grey line). (**C**) Internalisation of sulfo-NHS-SS-biotin and transferrin-647, for 15 min, in cells expressing AP180C-IRES-GFP. Internalised biotin was detected by MESNa treatment and labelling with streptavidin, a wash at pH3.0 removed external transferrin. Transfected cells are outlined in white. Bars are 20 µm. (**D**) Internalisation of sulfo-NHS-SS-biotin and transferrin-647, for 15 min, in cells expressing dynamin-2-K44A-dsRed. Internalised biotin was detected by MESNa treatment and labelling with streptavidin, a wash at pH3.0 removed external transferrin. Transfected cells are outlined in white. Bars are 20 µm. (**E**) Quantification of total protein and transferrin endocytosis in cells expressing AP180C-IRES-GFP as **C**. Each point is mean fluorescence intensity of one cell region, after background subtraction. Background was determined empirically from control experiments with only labelling at 4°C. Values are all normalised so mean of control = 1. Bars mean and SD, data are all from one experiment, the experiment was repeated three times. (**F**) Quantification is for cells expressing dynamin-2-K44A-dsRed as shown in **D**. See **E** for details.

The following figure supplements are available for figure 3:

**Figure supplement 1**. Correlation between streptavidin (total endocytosed protein) and transferrin intensities in endocytic vesicles.

**Figure supplement 2**. Effect of CTB-binding on transferrin intensities in endocytic vesicles.

---

directly. In order to ablate coated pit activity, cells were transfected with siRNA against the alpha adaptin subunit of the clathrin adaptor AP2 ('AP2 siRNA'). This blocked alpha-adaptin expression, and hence AP2 function (*Figure 4A*, *Figure 4B*) (*Motley et al., 2003*, *2006*). AP2 siRNA blocked both transferrin uptake and uptake of total biotinylated proteins to a similar extent (*Figure 4B*). SILAC (stable isotopic labelling by amino-acids in culture) followed by precipitation of surface biotinylated proteins with streptavidin-agarose provided a quantitative comparison of the relative abundance of plasma membrane proteins in control and AP2 siRNA treated cells. This revealed a clear and pronounced increase in the abundance of most proteins in AP2-siRNA cells (*Figure 4C* and *Figure 4—source data 1*). Analysis of SILAC ratios of the non-biotinylated cytosolic proteins present in the flow-through provided an internal control and did not reveal similar changes, confirming that the effects of AP2 depletion are largely restricted to the population of plasma membrane proteins (*Figure 4C*). This is consistent with the majority of plasma membrane proteins entering the cell via clathrin-coated pits.

The comparatively small list of proteins that are depleted from the plasma membrane upon AP2 siRNA treatment is heterogeneous (*Figure 4—source data 1*), and includes proteins thought to be internalised via clathrin-coated pits such as ADAM 10 (*Marcello et al., 2013*). The only obvious pattern within the list is the presence of most GPI-anchored proteins detected (*Figure 4D*). SILAC ratios for GPI-anchored proteins detected in the cytosolic flow-through were all close to one, so this is likely to reflect a depletion restricted only to the plasma membrane (*Figure 4—source data 1*). Flow cytometry to assay changes in plasma membrane levels of individual proteins confirmed that AP2 siRNA causes an accumulation of transferrin receptor at the plasma membrane, while levels of the GPI-anchored protein CD59 are reduced (*Figure 4—figure supplement 1*). As GPI-anchored proteins have been studied as potential cargoes

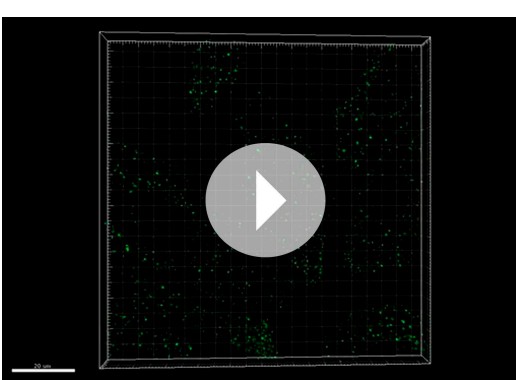

**Video 1**. Object recognition for quantification of cargo load in individual endocytic vesicles. Cells were labelled for 90 s at 37°C with sulfo-NHS-SS-biotin and transferrin-546. After MESNA treatment internalised proteins were labelled with streptavidin-488. 3D reconstructions were obtained from confocal z-stacks of whole cell volumes. Streptavidin channel (green), transferrin channel (red), an overlay of both channels, and then the same overlay with objects recognised as vesicles superimposed in blue, are shown consecutively.

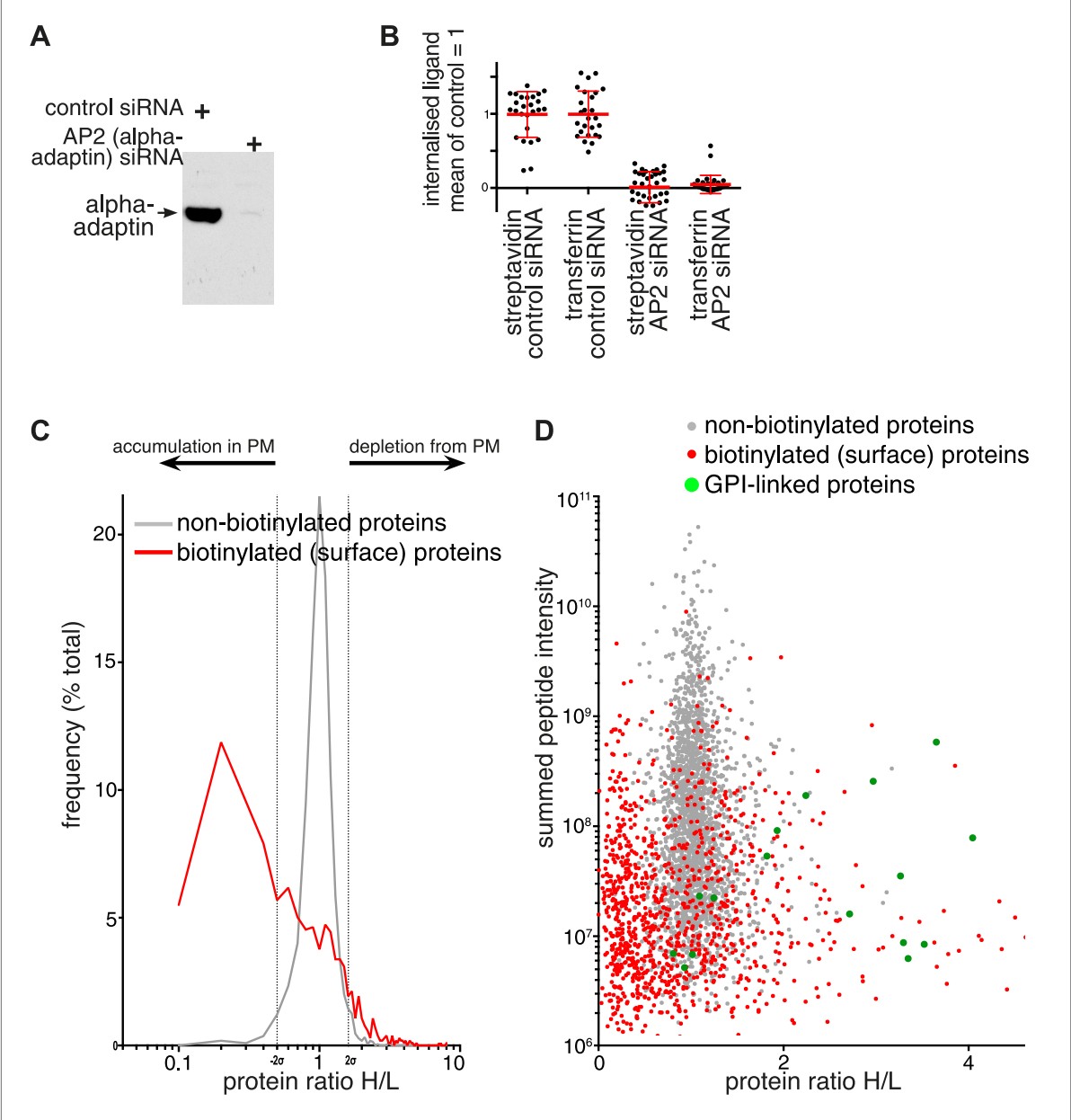

**Figure 4**. Changes in plasma membrane protein composition in cells depleted of AP2. (**A**) Western blot for the AP2 alpha subunit following non-targeting or alpha adaptin (AP2) siRNA treatment. (**B**) Internalisation of sulfo-NHS-SS-biotin and transferrin-647, for 15 min, in cells transfected with control and alpha adaptin (AP2) siRNAs. Internalised biotin was detected after MESNa treatment and labelling with streptavidin, a wash at pH3.0 removed external transferrin. Each point is one cell region, bars are mean and SD. Background was calculated from cells labelled at 4°C, and immunofluorescence identified those cells where the siRNA efficiently reduced alpha adaptin levels. (**C**) Frequency distribution of SILAC ratios for surface biotinylated (red line) and non-labelled (grey line) proteins from control (Heavy isotopes) and AP2-siRNA (Light isotopes) transfected cells. Dotted lines represent two standard deviations on either side of the mean for the SILAC ratio distribution of non-labelled proteins. (**D**) SILAC protein ratios comparing control (Heavy isotopes) and AP2-siRNA (Light isotopes) transfected cells, plotted against summed peptide intensities. Biotinylated plasma membrane proteins isolated by precipitation with streptavidin-agarose are shown in red. Non-biotinylated proteins corresponding to intracellular proteins are shown in grey and served as an internal control. GPI-anchored proteins, are shown in green.

The following source data and figure supplement is available for figure 4:

**Source data 1**. SILAC ratios for biotinylated and non-biotinylated proteins.

**Figure supplement 1**. Verification of changes in plasma membrane protein levels detected by SILAC.

for clathrin-independent endocytic pathways, we decided to focus further on this class of protein (*Nichols et al., 2001*; *Sabharanjak et al., 2002*; *Mayor and Riezman, 2004*).

## GPI-anchored proteins enter the cell via clathrin-coated pits

We produced plasmids for expression of SNAP-tagged versions of CD59, folate receptor and PrP, three different GPI-anchored proteins identified in our mass spectrometry data, as well a minimal SNAP-GPI construct. HeLa cells expressing each of the GPI-anchored proteins were incubated for 90 s at 37°C in the presence of BG-SS-488 to allow labelling and uptake, and extracellular fluorophore was removed by reduction with MESNa (*Figure 5A*). Uptake of BG-SS-488 was detected only in SNAP-expressing cells (*Figure 5A*). All four GPI-anchored proteins co-localised extensively with transferrin after internalisation (*Figure 5A*). When quantified, around 90% of all BG-SS-488 labelled, internalised GPI-anchored proteins co-localised with transferrin (*Figure 5B*). To assess the biological significance of the residual 10% we used precisely the same quantification method to analyse co-localisation between transferrin-alexa-647 and transferrin-alexa-546, after mixing and adding to cells for simultaneous internalisation. Again we detected around 90% co-localisation, so it is possible that the residual 10% can be accounted for by limitations in quantification rather than a biologically significant pool of GPI-anchored protein internalised separately from transferrin (*Figure 5B*).

Several previous studies, including some from our laboratory, have suggested that GPI-anchored proteins are endocytosed in a clathrin-independent manner (*Nichols et al., 2001*; *Nichols, 2002*; *Sabharanjak et al., 2002*; *Mayor and Riezman, 2004*; *Kirkham et al., 2005*; *Glebov et al., 2006*). We sought to explain this apparent discrepancy. In several of these studies CTB has been used as a marker for clathrin-independent endocytosis. As discussed previously, it was possible that CTB induces clathrin-independent uptake of GPI-anchored proteins. Object-based analysis of the transferrin load within BG-SS-488-positive vesicles in cells expressing SNAP-CD59 argues that this is not the case (*Figure 5—figure supplement 1*).

One type of experiment that can be interpreted as evidence for clathrin-independent endocytosis is the presence of endocytosed protein in vesicles separate from those labelled with transferrin (*Sabharanjak et al., 2002*; *Glebov et al., 2006*; *Bhagatji et al., 2009*). We did not observe this in the experiments outlined above (*Figure 5*), so we chose to pursue this further by conjugating a monoclonal antibody against CD59 to both alexa-546, and to SS-atto-488 (*Figure 6A*). Importantly, the disulfide bond in SS-atto-488 can be reduced with MESNa as previously. This meant we could follow uptake of endogenous CD59, and directly compare two different methods of assaying internalisation. Cells were labelled at 4°C with the doubly labelled antibody and transferrin, and then warmed to 37°C for 90 s. Subsequently cells were treated with PI-PLC to cleave GPI-anchors, and then with MESNa. The SS-atto-488 signal co-localised completely with transferrin. Alexa-546, however, was seen in many punctate structures that lack transferrin (*Figure 6A*). We interpret these data as strong evidence that the PI-PLC treatment did not remove all external GPI-anchored protein, while reduction with MESNa provided more stringent discrimination between internal and external pools of the antibody (*Figure 1A*, *Figure 1B*, *Figure 1—figure supplement 1*). Therefore, endogenous CD59 is indeed internalised in transferrin-containing vesicles, and apparent internalisation in vesicles lacking transferrin may, at least under the conditions we have employed in this study, be an artifact due to incomplete removal of surface bound antibody.

A second type of experiment used in many studies to provide evidence for uptake via clathrin-independent endocytosis is to perturb the function of clathrin-coated pits, and then assay continued endocytosis of a specific protein (*Doherty and McMahon, 2009*; *Hansen and Nichols, 2009*). We used overexpression of AP180-C to block formation of coated pits in a HeLa cell line stably expressing SNAP-CD59. In cells expressing AP180-C, endocytosis of SNAP-CD59 was blocked to the same extent as endocytosis of transferrin (*Figure 6B* and *Figure 6D*). Endocytosis of SNAP-CD59 was also blocked by moderate expression of dynamin-2-K44A (*Figure 6C* and *Figure 6E*). In cells expressing very high levels of the mutant dynamin, abundant macropinosomes could be detected, as well as smaller endocytic structures (*Figure 6—figure supplement 1*) (*Damke et al., 1995*). These were clearly induced by high levels of dynamin-2-K44A (*Figure 6—figure supplement 1*). Therefore moderate dynamin-2-K4AA expression does block the physiological mechanism that leads to internalisation of CD59 in unperturbed cells. As loss of both clathrin and dynamin function blocks uptake of CD59, the data imply that GPI-anchored proteins are likely to enter cells predominantly via clathrin-coated vesicles.

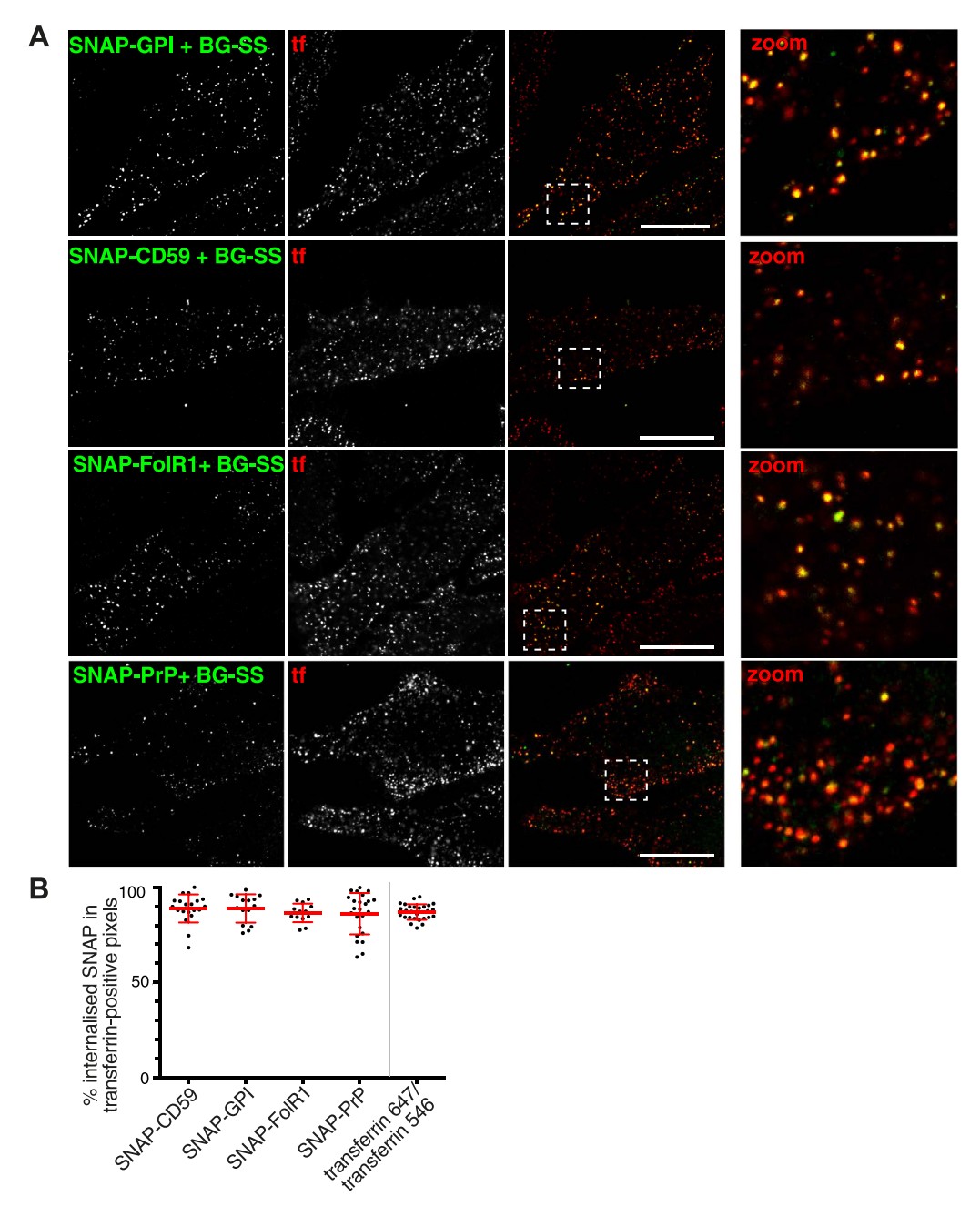

**Figure 5**. GPI-anchored proteins co-localise with transferrin in primary endocytic vesicles. (**A**) Confocal images of cells transfected with the SNAP-tagged GPI-anchored proteins indicated. Labelling with BG-SS-488 and transferrin-546 at 37°C for 90 s. External 488 fluorophore was removed by reduction with MESNa. Right hand panels are zoomed views of the regions indicated, bars are 10 μm. (**B**) Quantification of co-localisation between internalised GPI-anchored proteins revealed with BG-SS-488 and MESNa as in **A**, and transferrin-546. In order to provide an empirical estimate of the sensitivity of quantification, two fluorescently labelled transferrin probes were mixed and added to the cells. Bars are mean and SD.

The following figure supplement is available for figure 5:

**Figure supplement 1**. Effect of CTB-binding on transferrin intensities in endocytic vesicles defined by uptake of GPI-linked protein.

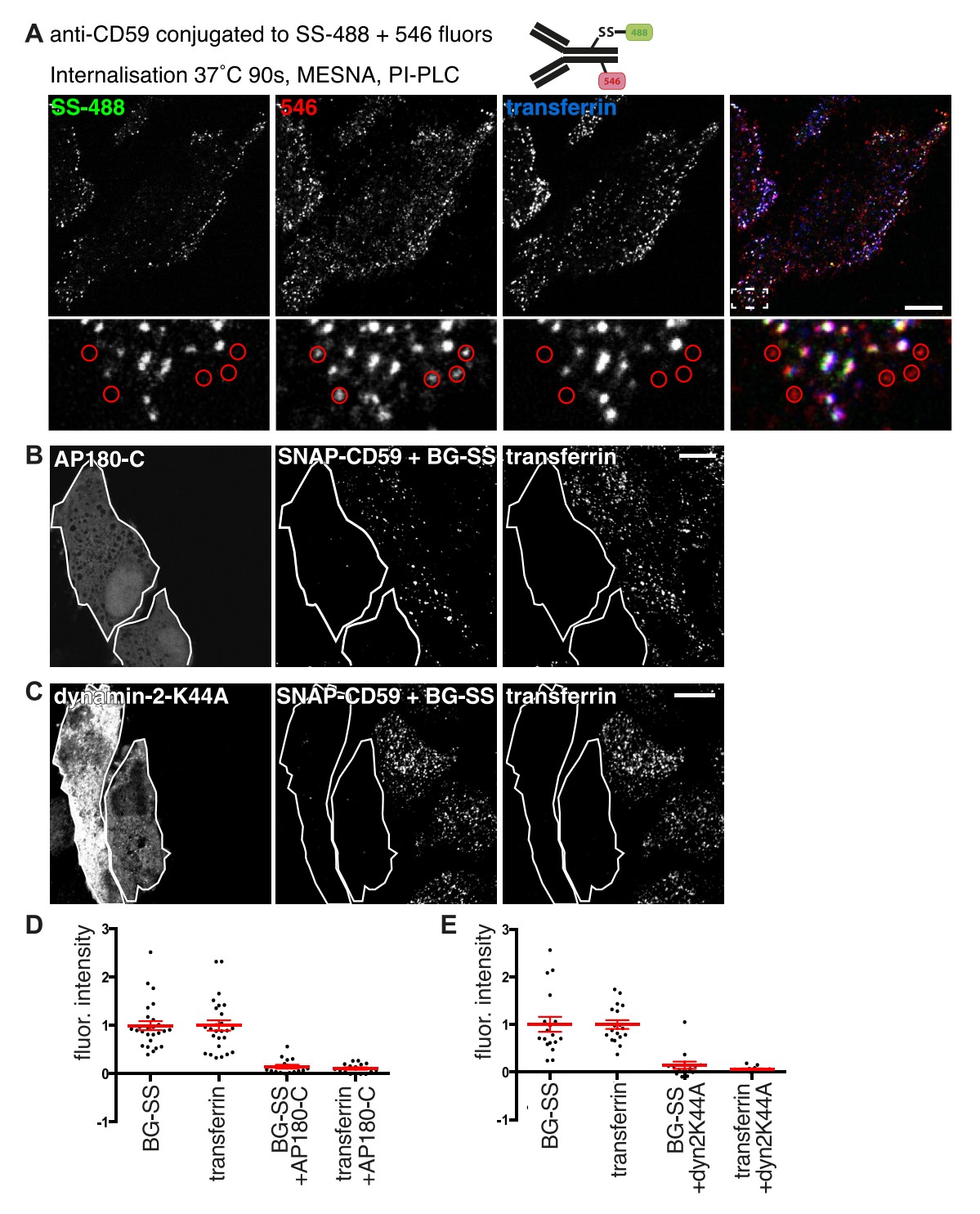

**Figure 6**. Clathrin-dependent endocytosis of GPI-anchored proteins. (**A**) Doubly labelled anti-CD59-546-SS-488 allows comparison of MESNa reduction and PI-PLC treatment as methods for detecting internalised GPI-anchored protein. Cells were labelled at 4°C, warmed to 37°C for 90 s, and treated consecutively with MESNa and PI-PLC. Circles indicated antibody-positive puncta that appear internalised, but are demonstrated to be extracellular by the absence of MESNa-protected 488 fluorophore. Bar is 10 μm. (**B**) Internalisation of BG-SS-549 and transferrin-647, for 15 min, in cells stably expressing SNAP-CD59 and transiently transfected with AP180C-IRES-GFP. Internalised BG-SS-549 was detected after MESNa treatment and wash at pH3.0 to remove external transferrin. Transfected cells are outlined in white. Bars are 10 μm. (**C**) Internalisation of BG-SS-488 and transferrin-647, for 15 min, in cells stably expressing SNAP-CD59 and transiently transfected with dynamin-2-K44A-dsRed. Internalised BG-SS-488 was detected by MESNa treatment, a wash
*Figure 6. Continued on next page*

*Figure 6. Continued*

at pH3.0 removed external transferrin. Transfected cells are outlined in white. Bars are 10 μm. (**D**) Quantification of SNAP-CD59 and transferrin endocytosis in cells expressing AP180C-IRES-GFP as **A**. Each point is mean fluorescence intensity of one cell region, after background subtraction. Background was determined empirically from control experiments with only labelling at 4°C. Values are all normalised so mean of control = 1. Bars mean and SEM, data are all from one experiment, the experiment was repeated three times. (**E**) Quantification is for cells expressing dynamin-2-K44A-dsRed as shown in **C**. See **D** for details.

The following figure supplement is available for figure 6:

**Figure supplement 1**. Endocytic structures induced by high dynamin-2-K44A expression.

## Labelling of the total population of endocytosed proteins and potential markers for clathrin-independent endocytosis

We used surface protein biotinylation with sulfo-NHS-SS-biotin to compare the distribution of total endocytosed protein with markers for possible clathrin-independent endocytic pathways. We examined co-localisation with flotillin 1 (***Glebov et al., 2006***; ***Stuermer, 2011***), caveolin 1 (***Rothberg et al., 1992***; ***Schneider et al., 2008***; ***Parton and Howes, 2010***), GRAF1-GFP (***Lundmark et al., 2008***), and ARF6-GFP (***Naslavsky et al., 2003***, ***2004***) after 90 s and 15 min of endocytosis at 37°C (***Figure 7***, ***Figure 7—figure supplement 1***). Co-localisation was quantified using a mask generated from the streptavidin channel as explained in ***Figure 2—figure supplement 3***. Importantly, as the different markers label abundant structures it was possible that low levels of co-localisation would be detected due to chance overlap rather than specific labelling of the same structures. To control for this, we repeated the co-localisation quantification with the two channels in all images offset from each other by 500 nm. This provides an empirical way to estimate the degree of overlap between channels that arises by chance.

In the case of caveolin 1, after both 90 s and 15 min some apparent co-localisation in punctate structures could be observed, and quantification confirmed that around 2% of caveolin 1 was present specifically in biotin-positive endosomes (***Figure 7A***, ***Figure 7B***, 15 min time-point images in ***Figure 7—figure supplement 1***). In the case of flotillin 1, although some plausible co-localisation could be observed in confocal images, quantification revealed that this may arise purely by chance as the same degree of pixel overlap was found in offset and original images (***Figure 7C***, ***Figure 7D***). In the case of GRAF1, internalised biotin was detected in GRAF1-GFP-positive puncta (***Figure 7E***). Quantification confirmed specific overlap between internalised biotin and GRAF1 (***Figure 7F***). The fact that our previous data show that most internalised biotin/total protein co-localises with transferrin made the strong prediction that these GRAF1-positive endosomes should also contain transferrin. This was indeed the case (***Figure 7E***). ARF6 showed a similar distribution to GRAF1, labelling transferrin-positive endosomes (***Figure 7G***, ***Figure 7H***). We conclude that GRAF1-GFP and ARF6-GFP are present on transferrin-positive endocytic vesicles likely to have arisen from budding of clathrin-coated pits from the plasma membrane, that flotillin 1 is not detected on early endosomes, and that caveolin 1 is present on a very small fraction of endosomes. All of these observations are consistent with our general conclusion, that at least 95% of endocytosed protein enters the cell via clathrin-coated pits.

## Distinct sorting mechanisms for transferrin receptor and GPI-anchored proteins lead to divergent effects on entry into coated pits

Our data argue that GPI-anchored proteins enter the cell via coated pits, but, they behave differently from many other clathrin cargoes when AP2 expression is suppressed over 5 days. There are several factors potentially involved in this effect, such as rates of synthesis, degradation and endocytic recycling of proteins with GPI anchors. We investigated further the factor most relevant to the focus of this study, sorting during endocytosis. The absence of a cytosolic domain means that GPI-anchored proteins are fundamentally different from high-affinity cargoes such as the transferrin receptor, as they can not be recruited to the nascent pit via direct recognition of endocytic sorting motifs by adaptor proteins. It is likely that GPI-anchored proteins are partially excluded from coated pits by steric crowding effects (***Bhagatji et al., 2009***). We reasoned that perturbing the recognition of endocytic sorting motifs could therefore cause differential effects on uptake of GPI-anchored proteins and transferrin receptor. As an initial test of this hypothesis, we used siRNA to partially reduce AP2 expression. AP2 recognises two major endocytic motifs (***Edeling et al., 2006***; ***Motley et al., 2006***; ***Kelly et al., 2008***; ***Kirchhausen et al., 2014***). We measured transferrin and SNAP-CD59 uptake by flow cytometry, 3 days after siRNA transfection.

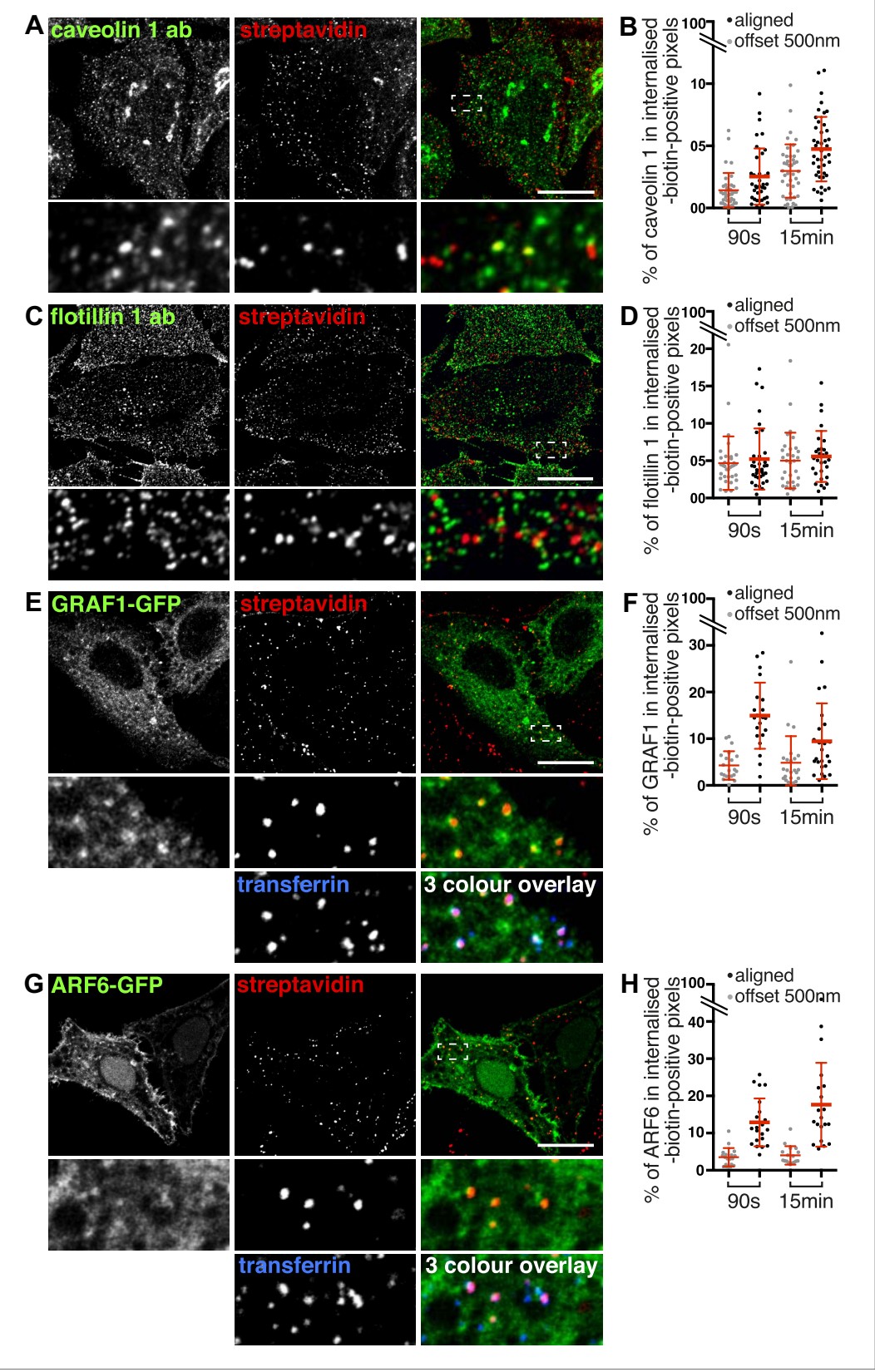

**Figure 7**. Labelling of the total population of endocytosed proteins does not provide evidence for significant protein flux through clathrin-independent pathways. (**A**, **C**, **E**, **G**) Confocal images showing distribution of the marker
*Figure 7. Continued on next page*

*Figure 7. Continued*

indicated (caveolin 1, flotillin 1, GRAF1, ARF6), together with total internalised protein after 90 s of endocytosis, revealed as in *Figure 1D*. In the case of GRAF1 and ARF6, additional images showing co-internalised transferrin are also shown. Bars are all 10 µm. (**B**, **D**, **F**, **H**) Quantification of co-localisation between the markers caveolin 1, flotillin 1, GRAF 1 and ARF6, and internalised protein labelled with sulfo-NHS-SS-biotin and MESNa treatment. Internalisation was for 90 s or 15 min. In order to establish empirically the degree of overlap between internalised protein and relevant marker expected by chance, quantification was carried out both with the images in the correct register, and also with one channel manually offset approximately 500 nm from the other. Bars are mean and SD, each point is one cell region.
The following figure supplement is available for figure 7:

**Figure supplement 1**. Labelling of the total population of endocytosed proteins does not provide evidence for significant protein flux through clathrin-independent endocytic pathways.

Cells still endocytosed significant amounts of both cargoes (*Figure 8—figure supplement 1*). We compared the ratio between the two cargoes, on a cell by cell basis, for the populations of control and AP2 siRNA transfected cells (*Figure 8A*). AP2 siRNA caused a more pronounced reduction of transferrin than CD59 uptake. Confocal imaging confirmed the flow cytometry results. In cells where AP2 levels were reduced, the reduction in transferrin uptake appeared more severe than reduction in SNAP-CD59 uptake (*Figure 8B*). Using these imaging data, we correlated the amount of AP2 remaining in each cell with the total fluorescence intensity of internalised transferrin and SNAP-CD59 (*Figure 8C*). The uptake of both cargoes was efficiently blocked when AP2 levels became close to zero, but transferrin uptake was clearly more sensitive to reduction of AP2 levels than uptake of SNAP-CD59. (*Figure 8C*; see also *Figure 4B*).

Specific amino acid changes in the YXXΦ binding site of the µ2 subunit of AP2 abolish binding of YXXΦ-containing cargoes like transferrin receptor (*Honing et al., 2005*; *Jackson et al., 2010*). Overexpression of this mutant form of µ2 for 4 days resulted in incorporation into endogenous AP2 complexes, and thereby provided a tool to further investigate sorting of transferrin and GPI-anchored proteins into coated pits (*Figure 8—figure supplement 2*). Overexpression of µ2(F174S/D176A), but not wild-type µ2, caused a dramatic differential effect on the uptake of the two cargoes after 90 s of labelling and internalisation. As predicted, transferrin uptake was reduced (*Honing et al., 2005*, *Motley et al., 2006*). Uptake of SNAP-CD59, however, was significantly increased (*Figure 8D*). Importantly, this divergence did not arise from separate endocytic structures, as internalised SNAP-CD59 still co-localised well with the residual internalised transferrin (*Figure 8E*). Moreover, when 3D object-based analysis was used to identify transferrin-positive vesicles, and to quantify the amount of SNAP-CD59 present in these vesicles, a clear increase in the SNAP-CD59 load in mutant-expressing cells was observed (*Figure 8F*). This analysis also confirmed the reduced transferrin load in endocytic vesicles in mutant-expressing cells (*Figure 8G*). These results are in agreement with our initial hypothesis of distinctive sorting of GPI-anchored proteins into clathrin coated pits. They argue that steric exclusion of GPI-anchored proteins from the nascent pit becomes less acute when recruitment of high-affinity cargoes by AP2, or potentially other adaptors is abolished (*Bhagatji et al., 2009*). More generally, the data imply that rates of endocytosis of high affinity cargoes may be more susceptible to a range of perturbations of coated pit function than cargoes like GPI-anchored proteins, which have a lower affinity for the nascent pit.

## Discussion

An imaging approach to studying endocytosis in unperturbed cells, where all surface proteins are labelled, is intrinsically easier to interpret than experiments where mutant proteins, siRNA, or chemical inhibitors cause loss of function, and potentially indirect or off-target effects, over days. The use of small, monovalent reducible chemical labels provides high temporal resolution, signal-to-noise ratio and efficiency of topological discrimination between intracellular and extracellular protein. This has allowed us to follow endocytosis of effectively all proteins, and thereby to observe all endocytic intermediates formed by the cell at specific time-points, starting with as little as 20 s of internalisation. Our results are at some variance with previous studies from our own and other laboratories, and with our preconceptions before being confronted with the data (*Hansen and Nichols, 2009*).

Our data support the conclusions that the predominant pathway for endocytosis in mammalian cells, accounting for at least 95% of total protein endocytic flux, is uptake via the clathrin-coated pit.

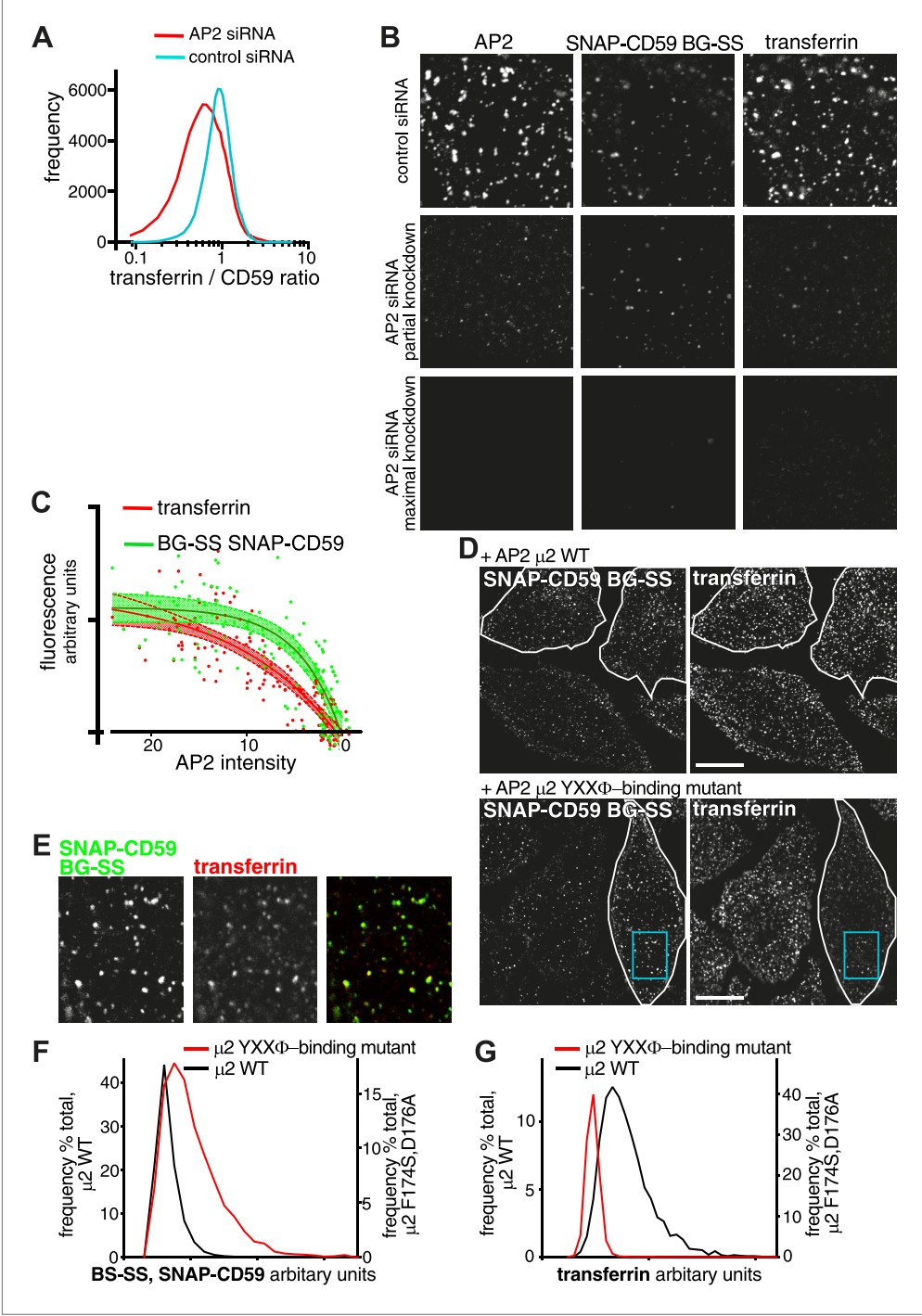

**Figure 8**. Differential effects on uptake of transferrin receptor and GPI-anchored proteins via coated pits. (**A**) Frequency distribution of the ratio between internalised transferrin and internalised SNAP-CD59 in individual HeLa cells, determined by flow cytometry as in **Figure 8—figure supplement 1**. (**B**) AP-2 knockdown can affect endocytosis of clathrin coated vesicle cargo proteins differentially. HeLa cells stably expressing SNAP-CD59 were transfected with AP2 siRNA and assayed at various timepoints up to 72 hr after transfection. Uptake of BG-SS-488 and transferrin-546 was for 15 min. Cells were MESNa treated, acid washed, fixed and stained with anti-alpha-adaptin. Top row; AP2 levels appear normal. Second row; AP2 levels are intermediate, transferrin uptake is blocked while SNAP-CD59 uptake is less severely inhibited. Third row; AP2 levels are very low and uptake of both SNAP-CD59 and transferrin is blocked. (**C**) Correlation of the amount of internalised transferrin and SNAP-CD59 with the

*Figure 8. Continued on next page*

*Figure 8. Continued*

amount of AP-2 present in each cell from the above experiment. Data were fit to a simple one-phase association. Shaded area around the curve fitted corresponds to 95% CI. (**D**) HeLa cells stably expressing SNAP-CD59 were transfected with µ2-IRES-GFP or µ2(F174S/D176A)-IRES-GFP. After 4 days, cells were incubated at 37°C for 150 s with transferrin-647 and BG-SS-546. The white lines outline transfected cells. The blue box highlights a region shown in **E**. Bars 20 µm. (**E**) Co-localisation between internalised SNAP-CD59 and transferrin in a cell expressing µ2(F174S/D176A)-IRES-GFP. (**F**) Frequency distribution of CD59 cargo load within individual vesicles. Cells were labelled as in **D**. Vesicles were identified as objects in 3D reconstructions from confocal images with Imaris software using the transferrin signal. (**G**) As **F**, but displaying transferrin cargo load in the same population of vesicles.

The following figure supplements are available for figure 8:

**Figure supplement 1**. Reduction of AP-2 (alpha adaptin) levels affects the amount of uptake of both transferrin and SNAP-CD59, 50 hr after siRNA transfection.

**Figure supplement 2**. Incorporation of mutant µ2 subunits into AP2 complexes.

---

Other mechanisms are not likely to make a significant contribution. Primary endocytic vesicles involved in clathrin-independent endocytosis should not contain significant amounts of transferrin receptor, and while we can detect a class of such vesicles they account for less than 5% of total endocytosis. Further data, including perturbations of clathrin-coated pit function, assay of the endocytosis of GPI-anchored proteins, and co-localisation experiments looking at candidate markers for clathrin-independent endocytic pathways are all consistent with this conclusion. A recent systematic analysis of multiple assays for endocytosis using siRNA screens also provides indirect support, in that it reveals extensive functional and regulatory links between uptake of cargoes previously thought to enter the cell via different mechanisms (*Liberali et al., 2014*). In addition, this study conspicuously does not detect new linked sets of proteins likely to act together in a concerted fashion to make novel types of endocytic vesicle. Some cargoes for clathrin coated pits have been reported to be present in sub-populations of early endocytic vesicles, most likely through mechanisms related to cargo-dependent recruitment of specific adaptors (*Lakadamyali et al., 2006*; *Mettlen et al., 2010*; *Henry et al., 2012*). However, there is good evidence that transferrin receptor does not behave in this way (*Lakadamyali et al., 2006*; *Kirchhausen et al., 2014*), and our analysis of transferrin load in primary endocytic vesicles confirms this. In the light of these observations, it is difficult to reconcile our data with reports of abundant clathrin-independent endocytic carriers or CLICs. We note that the ultrastructure of CLICs, although clearly different from clathrin-generated vesicles, is not dissimilar to that of the early endosomes that receive cargoes taken up via coated pits (*Harding et al., 1983*; *Willingham et al., 1984*; *Gruenberg et al., 1989*; *Hansen et al., 1991*; *Ullrich et al., 1996*).

Much of the literature on clathrin-independent endocytosis is based on perturbations with differential effects on the uptake of transferrin and putative clathrin-independent cargoes (*Puri et al., 2001*; *Sabharanjak et al., 2002*; *Sandvig et al., 2008*; *Doherty and McMahon, 2009*; *Hansen and Nichols, 2009*; *Lakshminarayan et al., 2014*). Our data suggest a possible re-interpretation of this type of experiment. We demonstrate that GPI-anchored proteins enter mammalian cells via clathrin coated pits, and that this is fully compatible with the differential effects on endocytic uptake between transferrin receptor and GPI-anchored proteins reported here and observed in previous studies (*Nichols et al., 2001*; *Sabharanjak et al., 2002*; *Naslavsky et al., 2004*; *Kumari and Mayor, 2008*). The coated pit is a crowded environment, and GPI-anchored proteins will have to compete for a place inside the nascent pit with cargo proteins possessing high affinity sorting signals recognised directly by adaptor proteins (*Kirchhausen et al., 2014*). When the protein levels of adaptors such as AP2 are reduced, or when the ability to bind and recruit certain cargoes is lost, it is likely that the pit becomes less crowded, thereby becomes more populated with low affinity or passive cargoes like GPI-anchored proteins, and yet retains some ability to bud. This leads to the differential effects on transferrin and CD59 uptake that we observed experimentally. Divergent effects of coated pit perturbation on cargoes which all demonstrably enter the cell via coated pits, means that such effects need no longer be interpreted as evidence for clathrin-independent endocytosis.

Our data place important constraints on models of endocytic flux. Additional endocytic mechanisms, like the still poorly understood set of protein machinery responsible for macropinocytosis, and the

budding of caveolae, contribute very little to total protein uptake but can be detected by our assays. Such clathrin-independent pathways could still be functionally important when uptake of specific cargoes, and the variety of cell types and functions in vivo are considered (*Commisso et al., 2013*; *Watanabe et al., 2013*). Nevertheless, we conclude that clathrin-independent pathways do not make a significant contribution to total endocytic flux in cultured cells.

## Materials and methods

### Cell culture and DNA transfections

HeLa, RPE and Cos7 cells where cultured at 37°C, 10% $CO_2$, in DMEM supplemented with 10% FBS. When required, cells were transfected with FugeneHD (Promega, Madison, WI), 14–20 hr before imaging.

### Constructs and cell lines

CD59 and FOLR1 were cloned from human cDNA. PrP was a gift from Manu Hegde. The N-terminal tagging of SNAP-CD59, SNAP-FOLR1 and SNAP-PrP was obtained by inserting the SNAPtag domain following the N-terminal signal peptide of each protein. The minimal SNAP-GPI was constructed by inserting a SNAPtag domain between the N-terminal signal peptide and the ω-2 site of human CD59. To obtain AP180C-IRES-GFP, AP180-C (A gift from H McMahon) was inserted into the multiple cloning site of IRES2-AcGFP1, using the XhoI and SacII sites. DynII (aa) K44A-dsRed was a gift from M Frick. GRAF1-GFP was a gift from H McMahon. AP2 μ2 and μ2(F174S/D176A) (Gifts from M Robinson) were cloned in IRES2-AcGFP1, using the XhoI and SacII sites. A typographic error for the YXXΦ binding mutant is present in previous publications (*Honing et al., 2005*; *Edeling et al., 2006*; *Motley et al., 2006*). The mutant used and characterised in those studies was as described here (David Owen, personal communication).

Following transfection, G418 (400 μg/ml) was used to select cells stably expressing SNAP-CD59. Resistant cells were sorted twice by flow cytometry to ensure retention of the construct and uniform expression levels.

### Antibodies and other reagents

antiCD59-546-ss488 was generated by conjugating anti-CD59 (MEM43) (Abcam, United Kingdom) simultaneously to NHS-ss-ATTO488 (ATTO-TEC, Germany) and NHS-AlexaFluor546 (Molecular Probes, Waltham, MA). Following conjugation the antibody was separated from unreacted reagents using size-exclusion chromatography. Antibodies and reagents were obtained from the following sources; AntiCD59-AlexaFluor647 (MEM43) (AbD Serotec, United Kingdom), mouse monoclonal (AP.6) against alpha adaptin used for immunofluorescence and immunoprecipitation (Abcam), mouse monoclonal against alpha adaptin used for western blots (BD, Franklin Lakes, NJ), rabbit polyclonal anti-caveolin 1 (BD), mouse monoclonal anti-flotillin 2 (BD), rabbit polyclonal against clathrin heavy chain (Abcam), Streptavidin-HRP (Pierce, Rockford, IL), PI-PLC from *Bacillus cereus* (Life Technologies, Waltham, MA), Streptavidin-488, -546 or -647 (Molecular Probes), Transferrin-546 or -647 (Molecular Probes), Cholera toxin subunit B (CTB) -647 (Molecular Probes), FM1-43FX (Molecular Probes), SNAP-surface 549 (NEB), BG-SS-488 and BG-SS-549 were kindly provided by our collaborators in NEB.

### siRNA transfection

Typically, non-targeting siRNA (Dharmacon, Lafayette, CO) or alpha-adaptin siRNA (Dharmacon) were delivered to the cells at a final concentration of 100 nM, using oligofectamine (Invitrogen). Transfections took place on days 1 and 3, while assays were carried out on day 5. For partial depletion of AP-2, one round of siRNA transfection took place and assays were performed at different timepoints up to 72 hr later. The siRNA targeting the alpha-subunit of AP-2, has been described previously (*Robinson et al., 2010*) [5'-GAG CAU GUG CAC GCU GGC CAdT dT-3'].

### Immunoprecipitation

AP2 complexes were immunoprecipitated with an anti-alpha adaptin antibody (AP.6) and protein G-sepharose after lysis with immunoprecipitation buffer (25 mM Tris–HCl pH 7.4, 150 mM NaCl, 1 mM EDTA, 1% Triton X-100 and 5% glycerol). To test for incorporation of the overexpressed mutant subunit into endogenous AP2 complexes, HeLa cells were transfected with μ2(F174S/D176A)-IRES-GFP and maintained in culture for the indicated periods.

## SILAC and mass spectrometry

HeLa cells were cultured for 7 days in R0K0 or R10K8 DMEM (Dundee Cell Products, United Kingdom) supplemented with dialysed fetal bovine serum (MWCO 10 kDa–Dundee Cell Products).

Following surface biotinylation, cells were lysed in 1% Triton X-100, 1% Octyl glucoside (Sigma, United Kingdom) in TBSE buffer (50 mM Tris pH 7.4, 150 mM NaCl, 5 mM EDTA) in the presence of protease inhibitors (Roche). The lysates were left to rotate in the coldroom for 30 min, and then spun at 20.000 rcf for 20 min. The supernatant was transferred to a clean eppendorf tube and incubated for 1 hr with high capacity streptavidin-agarose resin (Pierce). Every sample was then transferred to a chromatography column (Bio-Rad) and washed with 25 ml 1%Triton in TBSE. To elute biotinylated proteins the resin was incubated for 5 min with 100 mM DTT in TBS (50 mM Tris pH 7.4, 150 mM NaCl). SDS-PAGE gels were stained with Sypro Ruby (Lonza, Switzerland) or silver stain (Pierce). Peptide identification from each sample was done using LTQ Orbitrap XL (Thermo Scientific, Waltham, MA). Calculation of SILAC ratios and further data analysis were performed using MaxQuant (*Cox and Mann, 2008*) and Prism (GraphPad, San Diego, CA) respectively. The AP2 siRNA SILAC experiment was repeated three times, data shown are from one experiment. The same overall trend in terms of accumulation of most plasma membrane proteins in the AP2 siRNA treated cells and depletion of GPI-anchored proteins, were observed in all three experiments.

## Bioinformatic analysis of labelled plasma membrane proteins

A recently published estimate for protein copy numbers in HeLa cells (*Kulak et al., 2014*) was correlated with a list of human plasma membrane proteins [GO:0005886]. Plasma membrane abundance (PMA) for a protein x was calculated as shown;

$$PMA(x) = \frac{copy\ number\ of\ protein\ x}{sum\ copy\ number\ of\ all\ plasma\ membrane\ proteins} \times 100$$

We then compared this list of plasma membrane proteins with the list of biotinylated surface proteins we detected by mass spectrometry (*Figure 1—source data 1*).

## Surface biotinylation at 4°C

To suspend endocytosis, cells were moved to a coldroom (4°C) and washed twice with ice-cold PBS pH 7.9. Primary amines at the cell surface were labelled with 0.2 mg/ml sulfo-NHS-SS biotin in PBS pH 7.9. After 20 min, remaining sulfo-NHS-SS biotin was quenched with 50 mM Tris pH 8.0 in PBS, and the cells were washed two more times with PBS. Where required a further 10 min incubation at 4°C with PBS +12.5 µg/ml transferrin was carried out. To allow endocytosis, pre-warmed medium was added to the cells and the cultures were incubated at 37°C for various time points. Fluorescent transferrin, if required, was present in the prewarmed medium at 12.5 µg/ml. At the same time, positive control cells to estimate the amount of surface labelling, and negative control cells to estimate the efficiency of reduction would remain in the coldroom.

Following internalisation, cells were chilled with pre-cooled PBS to immediately arrest endocytosis and then washed with ice-cold MESNa buffer without BSA in the coldroom. Remaining surface exposed label was removed by incubating the cells 2 × 10 min in cold 100 mM MESNa in MESNa buffer (MESNa buffer: 50 mM Tris, 100 mM NaCl, 1 mM EDTA, 0.2wt/vol BSA, pH 8.6) Finally, transferrin, if used, was removed from the plasma membrane by acid wash at 4°C (2 × 2 min with 150 mM Glycine, pH 3.0).

## Surface biotinylation at 37°C

The above protocol was modified for labelling at 37°C, with no labelling step at 4°C preceding the timeframe for endocytosis. Sulfo-NHS-SS Biotin was dissolved in pre-warmed HBSS (HBSS + calcium, magnesium, glucose—Gibco) at a final concentration of 0.2 mg/ml. If required fluorescent transferrin was added, at a final concentration of 12.5 µg/ml, before addition to the cells. After incubation at 37°C for the indicated time cells were rapidly chilled using pre-cooled PBS at 4°C, and moved to the coldroom to remove surface exposed biotin or transferrin as described above.

## Endocytic assay using the reducible SNAP-tag

BG-SS-fluorophore and fluorescent transferrin were diluted in prewarmed DMEM supplemented with 10%FBS (Final concentrations; BG-SS-Fluorophore 2.5 nmol/ml, transferrin 12.5 µg/ml). After incubation

at 37°C for the indicated time, cells were rapidly chilled using pre-cooled PBS at 4°C, and moved to the coldroom to arrest endocytosis and remove surface exposed label or transferrin as described previously.

### Detection of endocytic vesicles using the amphiphilic dye FM1-43FX

The protocol used has been previously described (*Diefenbach et al., 1999*). Briefly, cells were labelled for 1 min at 4°C with 10 µM FM1-43FX (Invitrogen) in HBSS. Following internalisation at 37°C, excess FM1-43FX was removed from the plasma membrane by incubating the cells with PBS without FM1-43FX for 20 min at 4°C. Cells were imaged at 4°C without fixation. Temperature was maintained by addition of frozen, crushed DMEM to the cell chambers.

### PI-PLC treatment

GPI-anchored proteins were removed from the plasma membrane by incubating the cells at 4°C with PI-PLC (10 U/ml) in PBS for 20 min.

### Flow cytometry

For flow cytometry, endocytic assays were performed as described above. Cells were then trypsinised, resuspended in cold 0.2% FBS in PBS and analysed using BD LSRFortessa.

### Fluorescence microscopy

For microscopy, imaging dishes (CG 1.5–Miltenyi Biotec) were coated with fibronectin (5 µg/ml in PBS) (Sigma Aldrich) overnight. The following day cells were seeded at a density of $6 \times 10^3$ cells/cm$^2$.

After the internalisation assays, cells were fixed and permeabilised with either 4% paraformaldehyde in PBS followed by 0.1% Saponin or with methanol at −20°C (required for staining with anti-flotillin antibodies). Images were acquired on a Zeiss 780 confocal microscope using a 63× 1.40NA Plan-Apochromat objective (Zeiss) and a GaAsP detector.

### Image processing and quantification

Images were processed in Image J. All images were subjected to noise reduction with Gaussian blur σ = 0.7, and contrast settings were adjusted for optimal visualisation of colour overlays. The black level in all images was set using levels empirically determined from negative control experiments to exclude cellular auto-fluorescence and other non-specific sources of signal. Quantification of co-localisation is explained in *Figure 2—figure supplement 3*, and the relevant legend. In brief, one channel of a 2-colour image was converted into a binary mask that could be used to isolate those pixels from the second channel that are positive in the mask. Quantification of mean fluorescence intensity was carried out by simply drawing cell outlines as regions of interest in Image J. Importantly, empirically determined background from negative control experiments in which cells were incubated with labels only at 4°C was subtracted from all values. As there is inherent variability in both signal and background from cell to cell, this resulted in negative values in some instances.

High resolution confocal stacks (150 nm interval) were used for volume rendering and object identification with Imaris (Bitplane). Primary endocytic vesicles were detected in the streptavidin channel using a region growing algorithm that used intensity parameters (mean, center, maximal, standard deviation) as detection criteria. To estimate and account for noise and random overlap we offset the transferrin channel by 20 voxel in x (~500 nm). We then compared the updated statistics for individual endosomes with the values for the same endosomes when the two channels were correctly aligned.

## Acknowledgements

We thank Mark Skehel and the LMB Mass Spec. team for analysis of SILAC experiments. Sean Munro and Ramanujan Hegde provided insightful comments on the manuscript.

## Additional information

### Competing interests

IRC: An employee of New England Biolabs Inc. New England Biolabs Inc. has a commercial interest in successful application of reagents used in this study. The other authors declare that no competing interests exist.

## Funding

| Funder | Author |
| --- | --- |
| Medical Research Council | Vassilis Bitsikas |
| Medical Research Council | Benjamin J Nichols |

The funder had no role in study design, data collection and interpretation, or the decision to submit the work for publication.

## Author contributions

VB, Conception and design, Acquisition of data, Analysis and interpretation of data, Drafting or revising the article; IRC, Drafting or revising the article, Contributed unpublished essential data or reagents; BJN, Conception and design, Analysis and interpretation of data, Drafting or revising the article

## Author ORCIDs

Vassilis Bitsikas, http://orcid.org/0000-0002-1731-9910

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
