## [Decision Letter]

Thank you for sending your work entitled *“*clathrin-independent pathways do not contribute significantly to endocytic flux” for consideration at *eLife.* Your article has been favorably evaluated by Randy Schekman (Senior editor) and 3 reviewers, one of whom (Suzanne Pfeffer) is a member of our Board of Reviewing Editors.

The Reviewing editor and the other reviewers discussed their comments before we reached this decision, and the Reviewing editor has assembled the following comments to help you prepare a revised submission.

This study is aimed at rigorously testing whether clathrin-independent endocytosis operates as a sizable parallel uptake route to clathrin-mediated endocytosis. The Introduction section frames the problem beautifully. After settling on reducible surface biotinylation for a labeling procedure, it was confirmed that this tags bulk surface constituents. Individual cargo at the surface could be followed using a reducible fluorescent ligand for the SNAP tag. Using this approach, >95% of internalized biotinylated proteins colocalized with endocytosed transferrin (Tf) after 1-2 min at 37°C. A pulse of NHS-SS-biotin at 37°C for ∼1-2 min together with Tf also produced a high degree of overlap between Tf and streptavidin signals. The remainder of the manuscript reinforces this principal finding. The authors conclude that clathrin-mediated endocytosis represents the major route for internalization for the plasma membrane and that cargo packaging is subject to steric competition.

This is a systematic, carefully controlled, well interpreted and thought provoking study that forces serious reconsideration of a large body of previously published literature. The data are generally compelling and formally question the quantitative significance of clathrin-independent endocytosis, at least in the cell types examined.

The most paradoxical results are with regard to the CLIC/GEEC pathway, which through the work of Parton and Mayor has been suggested to account for the majority of bulk/fluid phase uptake in cells. The results from two papers in particular (Kirkham et al, 2005 (ref 17); Howes et al, 2010, which is not, but should be cited) need to be reconciled with the current findings. In these studies Parton and colleagues use quantitative electron microscopy studies with DAB reactions and surface quenching with ascorbic acid to also look at very early endocytic intermediates and identify CLIC, which have distinct morphologies and lack Tfn. Their morphometric analysis suggests that they account for the majority of bulk uptake in the cells they examined. We see two possibilities for reconciling these results:

1) The lack of sensitivity of detection of nascent clathrin-independent endocytic vesicles. CME is a concentrative uptake mechanism so that B-cargo and Tfn will be ∼20-fold more concentrated in nascent CCV derived vesicles as compared to bulk PM proteins in CLIC. While the data in Figure 3—figure supplement 1 appears to argue that even vesicles with very low levels of streptavidin contain Tfn it would be important somehow ensure that diffraction limited vesicles with the same label intensity as what would be expected for bulk membrane also are enriched in Tfn. That is there should be 20-fold more Tfn than bulk membrane in these nascent vesicles. Could this be done with a bulk membrane marker such as an FM143 dye? The GPI-probes should show a >20:1 enrichment of Tfn vs GPI. Do these intensity differences match these expectations or are the authors unable to detect poorly labeled nascent CLICs?

2) Another possible difference is if cholera toxin (Ctx), which is used as a CLIC marker by Parton et al, actually induces formation of CLICs. This is a very easy experiment to do, namely to repeat the experiments in Figures 2 and 5 in the presence of unlabeled Ctx. We feel this will greatly strengthen the current story.

A second set of experiments not reconciled by the data are results of the effects of dominant-negative dynamin mutants and Dyn2 null cells. Damke et al., 1995 showed that bulk-endocytosis is restored in cells expressing ts-dynamin-1 within 30 minutes after shift to the nonpermissive temperature. This compensatory, clathrin-independent endocytosis occurred through nascent vesicles smaller than CCVs as judged by HRP uptake and EM studies. Liu et al., 2008 showed that dynamin-2 null cells continue to take up cholera toxin and Bodipy-LacSer even though transferrin uptake and macropinocytosis were inhibited. Can the authors show induction of clathrin-independent pathways other than macropinocytosis? Please discuss.

The next comments are listed below in the spirit of making the manuscript more complete and in allowing the non-specialist reader to fully comprehend the results and their significance.

1) The GPI-anchored proteins stood out in the MS analysis as they were depleted, rather than enriched, at the cell surface in an AP-2-silenced background. The protein H/L ratios of the green-labeled GPI-anchored proteins in Figure 4 shows that these biotinylated forms of GPI-anchored proteins are clearly lower on the surface of AP-2 knocked-down cells. However, does the manuscript’s whole protein data set show whether GPI-anchored proteins accumulate inside the AP-2 siRNA cell population rather than being extensively degraded? The underlying mechanistic basis for the surface loss is not really commented on directly or rationalized. It seems to me that they should explain why this occurs. GPI-anchored proteins do recycle back to the surface more slowly, but published data indicate this is due to interaction with cholesterol in the endosomal compartment. After several days, AP-2 knocked-down cells should be relatively cholesterol deficient due to LDL receptor accumulation on the cell surface and extremely limited endocytic flow. Do the GPI-anchored proteins in the intracellular fraction of AP-2-depleted cells recycle? If not, does this argue for a requirement for plasma membrane constituents, stalled at the cell surface in AP-2 knocked-down cells, for GPI-anchored protein recycling?

2) One of the primary ways that the CIE pathway has been discriminated from the clathrin-mediated pathway is with Arf6. In this work, Arf6 is hardly mentioned. Perhaps the authors would like to comment on the possible role of Arf6 in the events they are documenting.

3) A key set of mechanistically-revealing experiments are presented in Figure 8, were uncoupling between Tf and SNAP-CD59 as a function of AP-2 concentration is reported. The difference between the extensively and moderately AP-2-depleted cells seems very small compared with the level of AP-2 in the 'normal' cell (panel C). The white- and blue-outlined cells both have very little AP-2. It is not stated clearly what time point these cells were analyzed after AP-2 siRNA application, nor for the quantitative data in panel D. It is difficult to follow the extent of AP-2 depletion in these cells at a comparative biochemical level. Is this 'uncoupling' effect only noted when AP-2 levels are very low? The graph in panel D suggests this is not the case. Perhaps a more representative set of images could be shown?

4) To our knowledge, most experiments using overexpressed mu2 subunits have been disappointing. AP-2 has a half-life of >30 hours in cultured cells. There is no direct evidence provided that the mutant mu2 protein overexpressed is actually incorporated into functional AP-2 tetramers.

---

## [Author Response]

We have carried out several new experiments to address the issues raised by reviewers. I am confident that the new experiments have improved the manuscript. A full discussion of each of the points made by reviewers, and a description of the new experiments, follows below.

The most significant new experiments are: 1) Repeating the analysis of cargo load of endocytic vesicles with and without cholera toxin B subunit (CTB), to control for possible effects of CTB-binding. 2) Imaging and measuring co-localisation between transferrin and a label for endocytosed membrane, FM-143FX, to control for the (in our opinion remote) possibility of endocytic vesicles that are not detected by labelling of all plasma membrane proteins with sulfo-NHS-SS-biotin.

There are also additional new experiments, including showing that the AP2 mu2 subunit mutants that we use are indeed incorporated into AP2 complexes, and analysing co-localisation between ARF6 and endocytosed proteins.

The most paradoxical results are with regard to the CLIC/GEEC pathway, which through the work of Parton and Mayor has been suggested to account for the majority of bulk/fluid phase uptake in cells. The results from two papers in particular (Kirkham et al, 2005 (ref 17); Howes et al, 2010, which is not, but should be cited) need to be reconciled with the current findings. In these studies Parton and colleagues use quantitative electron microscopy studies with DAB reactions and surface quenching with ascorbic acid to also look at very early endocytic intermediates and identify CLIC, which have distinct morphologies and lack Tfn. Their morphometric analysis suggests that they account for the majority of bulk uptake in the cells they examined. We see two possibilities for reconciling these results:

*1) The lack of sensitivity of detection of nascent clathrin-independent endocytic vesicles. CME is a concentrative uptake mechanism so that B-cargo and Tfn will be ∼20-fold more concentrated in nascent CCV derived vesicles as compared to bulk PM proteins in CLIC. While the data in*
Figure 3—figure supplement 1
*appears to argue that even vesicles with very low levels of streptavidin contain Tfn it would be important somehow ensure that diffraction limited vesicles with the same label intensity as what would be expected for bulk membrane also are enriched in Tfn. That is there should be 20-fold more Tfn than bulk membrane in these nascent vesicles. Could this be done with a bulk membrane marker such as an FM143 dye? The GPI-probes should show a >20:1 enrichment of Tfn vs GPI. Do these intensity differences match these expectations or are the authors unable to detect poorly labeled nascent CLICs?*

This point relates to the possibility that our reducible biotin + streptavidin labelling protocol do not detect all endocytic structures. This is of course a valid concern. We point out, however, that as we are labelling essentially all proteins, any putative vesicles not detected in our protocol would have to be either much smaller than clathrin coated vesicles, or highly depleted of protein relative to such vesicles. In this context it is important to emphasise that, as shown in Figure 3—figure supplement 1, there is no correlation between the biotin intensity of a vesicle and the probability that we fail to detect transferrin in it; it is not the case that low biotin intensity vesicles close to the detection threshold are significantly less likely to contain transferrin than higher biotin intensity vesicles. Additionally, macropinosomes (which are likely to have a similar membrane protein composition to the plasma membrane) are readily detected using our approach, as are small and scarce caveolar vesicles (Figure 2—figure supplement 1 and Figure 7). The structures described as CLICs by electron microscopy are, in fact, larger than clathrin-coated vesicles [1,2], so all in all it seems unlikely that our approach would fail to detect them.

We have devised additional experiments to support these arguments. The central question is whether there are membrane vesicles that both lack transferrin and are not detected with our reducible biotin + streptavidin protocol. We labeled cells with the styryl membrane dye FM1-43FX, along with transferrin, and allowed uptake for 90s at 37°C. There was effectively complete co-localisation between the FM1-43FX in small endocytic vesicles and transferrin, and as expected FM1-43FX also labeled obviously larger and relatively infrequent macropinosomes. This point is now made in the text, and representative images and quantification are included as Figure 2—figure supplement 7. These new data help to rule out the possibility of abundant membrane vesicles that do not contain transferrin.

*2) Another possible difference is if cholera toxin (Ctx), which is used as a CLIC marker by Parton et al, actually induces formation of CLICs. This is a very easy experiment to do, namely to repeat the experiments in*
Figures 2 and 5
*in the presence of unlabeled Ctx. We feel this will greatly strengthen the current story*.

This is a very good suggestion. We have done precisely what the reviewer suggests and analysed the transferrin cargo load in biotin- or SNAP-CD59-positive vesicles + / - Ctx (which we refer to as CTB in the manuscript). The data, in Figure 3—figure supplement 2, and Figure 5—figure supplement 1, show that Ctx does not induce a class of vesicles which contain biotin or SNAP-CD59 but not transferrin.

This still leaves the issue of relating our data with the papers identifying CLICs somewhat unresolved [1,2]. CLICs, like macropinosomes, could be present, or up-regulated, in specific circumstances such as at the leading edge of migrating cells. We have looked at RPE cells, which spontaneously migrate in culture, and did not see evidence for this (Figure 2—figure supplement 5). Another possibility is that many structures identified by ultrastructure as CLICs may, in fact, be transferrin-positive early endosomes. The various EM studies on the morphology of early endosomes under different conditions yield examples of early endosome structures that look like CLICs, but contain transferrin or other clathrin cargoes (compare [2]-Figure 4A, with [3]-Figure 6F, [4]-Figure 1, [5]-Figure 9, [6]-Figures 8,10, [7]-Figure 1 *etc*.). Therefore we suggest that it is hard to define CLICs by a specific morphology, and reliance on transferrin-HRP or CTB-HRP as markers means that simultaneous detection of both conventional early endosomes and potential CLICs in the same EM sections is difficult [2]. We have added a brief section in our discussion where we try to reconcile the results in the literature by pointing out the morphological similarities between CLICs and early endosomes.

*A second set of experiments not reconciled by the data are results of the effects of dominant-negative dynamin mutants and Dyn2 null cells. Damke et al., 1995 showed that bulk-endocytosis is restored in cells expressing ts-dynamin-1 within 30 minutes after shift to the nonpermissive temperature. This compensatory, clathrin-independent endocytosis occurred through nascent vesicles smaller than CCVs as judged by HRP uptake and EM studies. Liu et al., 2008 showed that dynamin-2 null cells continue to take up cholera toxin and Bodipy-LacSer even though transferrin uptake and macropinocytosis were inhibited. Can the authors show induction of clathrin-independent pathways other than macropinocytosis? Please discuss*.

Again, this point is well taken. On reflection, it was incorrect of us to describe the endocytosis induced by the dynamin 2 K44A mutant as macropinocytosis. Close inspection of the image we show, Figure 6—figure supplement 1, suggests that biotin-positive, transferrin-negative structures with a range of sizes potentially including sub-resolution vesicles as well as larger vesicles, are induced by the mutant. Given the complex mode of action of dynamin, interpretation of the effects of overexpression of the K44A mutant in terms of precise mechanisms is difficult [8]. The important finding from the point of view of our argument is that these structures are indeed clearly induced by the mutant, as demonstrated both by the image and by flow cytometry (Figure 6—figure supplement 1). We have amended the text accordingly. The finding that dynamin 2 K44A induces endocytic events that usually do not happen / are relatively infrequent highlights what we see as one of the main advantages of the approach used in our paper, namely analysis of the identity of endocytic carriers in unperturbed cells.

*The next comments are listed below in the spirit of making the manuscript more complete and in allowing the non-specialist reader to fully comprehend the results and their significance*.

*1) The GPI-anchored proteins stood out in the MS analysis as they were depleted, rather than enriched, at the cell surface in an AP-2-silenced background. The protein H/L ratios of the green-labeled GPI-anchored proteins in*
Figure 4
*shows that these biotinylated forms of GPI-anchored proteins are clearly lower on the surface of AP-2 knocked-down cells. However, does the manuscripts whole protein data set show whether GPI-anchored proteins accumulate inside the AP-2 siRNA cell population rather than being extensively degraded? The underlying mechanistic basis for the surface loss is not really commented on directly or rationalized. It seems to me that they should explain why this occurs. GPI-anchored proteins do recycle back to the surface more slowly, but published data indicate this is due to interaction with cholesterol in the endosomal compartment. After several days, AP-2 knocked-down cells should be relatively cholesterol deficient due to LDL receptor accumulation on the cell surface and extremely limited endocytic flow. Do the GPI-anchored proteins in the intracellular fraction of AP-2-depleted cells recycle? If not, does this argue for a requirement for plasma membrane constituents, stalled at the cell surface in AP-2 knocked-down cells, for GPI-anchored protein recycling?*

We wrote the original manuscript deliberately trying to avoid claiming that our data on sorting of GPI-linked proteins into clathrin coated pits provide a complete explanation for the depletion of GPI-linked proteins from the plasma membrane following AP2 depletion. We fully agree with the reviewer that other factors including rates of recycling from intracellular compartments, rates of synthesis / Golgi exit, and changes in cellular lipid composition, could also be relevant. A full examination of these possibilities is clearly beyond the scope of the current manuscript. The SILAC data is presented in the current manuscript to substantiate the point that most proteins accumulate in the membrane in the absence of AP2, but GPI-linked proteins behave differently and therefore their sorting behaviour requires further study. Our subsequent experiments relate to endocytosis and sorting of GPI-linked proteins, the issue relevant to the paper as a whole.

In order to clarify the text we now mention some of the additional factors listed above that could be relevant to depletion of GPI-linked proteins in the plasma membrane. Also, the SILAC MS does indeed indicate that the intracellular pool of GPI-linked proteins is unchanged when AP2 is depleted, as GPI-anchored proteins are detected in the non-biotinylated protein flow-through fraction, and the SILAC ratios for these proteins are close to 1. Again, this point is now made explicitly in the text.

*2) One of the primary ways that the CIE pathway has been discriminated from the clathrin-mediated pathway is with Arf6. In this work, Arf6 is hardly mentioned. Perhaps the authors would like to comment on the possible role of Arf6 in the events they are documenting*.

We now include ARF6-GFP in the survey of potential markers for clathrin-independent endocytic pathways in Figure 7. Like GRAF1, ARF6 is detected on transferrin-positive endosomes but not, in our hands, on transferrin-negative structures.

*3) A key set of mechanistically-revealing experiments are presented in*
Figure 8*, were uncoupling between Tf and SNAP-CD59 as a function of AP-2 concentration is reported. The difference between the extensively and moderately AP-2-depleted cells seems very small compared with the level of AP-2 in the 'normal' cell (panel C). The white- and blue-outlined cells both have very little AP-2. It is not stated clearly what time point these cells were analyzed after AP-2 siRNA application, nor for the quantitative data in panel D. It is difficult to follow the extent of AP-2 depletion in these cells at a comparative biochemical level. Is this 'uncoupling' effect only noted when AP-2 levels are very low? The graph in panel D suggests this is not the case. Perhaps a more representative set of images could be shown?*

As suggested by the reviewer we have improved the presentation of the data in Figure 8. This experiment forms Figure 8 in the new manuscript. We have increased the size of the images and chosen new examples that make the point more clearly.

*4) To our knowledge, most experiments using overexpressed mu2 subunits have been disappointing. AP-2 has a half-life of >30 hours in cultured cells. There is no direct evidence provided that the mutant mu2 protein overexpressed is actually incorporated into functional AP-2 tetramers*.

This is a good point. The long half-life of AP-2 complexes suggests that overexpression of mutant subunits needs to be for several days before the exogenously expressed subunits get incorporated into endogenous complexes. The effects shown in Figure 8 were observed after 4 days of overexpression of the mutant subunit. We have now carried out co-IP experiments to show that mutant ∝2 gets incorporated into endogenous complexes, and this incorporation is dependent on the time of overexpression (Figure 8—figure supplement 2). We hope the new data will improve the manuscript.Howes MT, Kirkham M, Riches J, Cortese K, Walser PJ, et al. (2010) Clathrin-independent carriers form a high capacity endocytic sorting system at the leading edge of migrating cells. *J Cell Biol*
**190**: 675-691. doi: 10.1083/jcb.201002119. Epub 201002010 Aug 201002116.Kirkham M, Fujita A, Chadda R, Nixon SJ, Kurzchalia TV, et al. (2005) Ultrastructural identification of uncoated caveolin-independent early endocytic vehicles. *J Cell Biol*
**168**: 465-476.Ullrich O, Reinsch S, Urbe S, Zerial M, Parton RG (1996) Rab11 regulates recycling through the pericentriolar recycling endosome. *J Cell Biol*
**135**: 913-924.Willingham MC, Hanover JA, Dickson RB, Pastan I (1984) Morphologic characterization of the pathway of transferrin endocytosis and recycling in human KB cells. *Proc Natl Acad Sci U S A*
**81**: 175-179.Hansen SH, Sandvig K, van Deurs B (1991) The preendosomal compartment comprises distinct coated and noncoated endocytic vesicle populations. *J Cell Biol*
**113**: 731-741.Harding C, Heuser J, Stahl P (1983) Receptor-mediated endocytosis of transferrin and recycling of the transferrin receptor in rat reticulocytes. *J Cell Biol*
**97**: 329-339.Gruenberg J, Griffiths G, Howell KE (1989) Characterization of the early endosome and putative endocytic carrier vesicles in vivo and with an assay of vesicle fusion in vitro. *J Cell Biol*
**108**: 1301-1316.Loerke D, Mettlen M, Yarar D, Jaqaman K, Jaqaman H, et al. (2009) Cargo and dynamin regulate clathrin-coated pit maturation. *PLoS Biol*
**7**: e57. doi: 10.1371/journal.pbio.1000057.